# Memory Caching: RNNs with Growing Memory

**Ali Behrouz** [1]  **Zeman Li** [1]  **Yuan Deng** [1]  **Peilin Zhong** [1]  **Meisam Razaviyayn** [1]  **Vahab Mirrokni** [1]

## Abstract

Transformers have been established as the de-facto backbones for most recent advances in sequence modeling, mainly due to their growing memory capacity that scales with the context length. While plausible for retrieval tasks, it causes quadratic complexity and so has motivated recent studies to explore viable subquadratic recurrent alternatives. Despite showing promising preliminary results in diverse domains, such recurrent architectures underperform Transformers in recall-intensive tasks, often attributed to their fixed-size memory. In this paper, we introduce Memory Caching (MC), a simple yet effective technique that enhances recurrent models by caching checkpoints of their memory states (a.k.a. hidden states). MC allows the effective memory capacity of RNNs to grow with sequence length, offering a flexible trade-off that interpolates between the fixed memory (i.e., $\mathcal{O}(L)$ complexity) of RNNs and the growing memory (i.e., $\mathcal{O}(L^2)$ complexity) of Transformers. We propose four variants of MC, including gated aggregation and sparse selective mechanisms, and discuss their implications on both linear and deep memory modules. Our experimental results on language modeling, and long-context understanding tasks show that MC enhances the performance of recurrent models, supporting its effectiveness. The results of in-context recall tasks indicate that while Transformers achieve the best accuracy, our MC variants show competitive performance, close the gap with Transformers, and performs better than state-of-the-art recurrent models.

[1]Google Research, United States. Correspondence to: Ali Behrouz <alibehrouz@google.com>.

*Proceedings of the 43rd International Conference on Machine Learning*, Seoul, South Korea. PMLR 306, 2026. Copyright 2026 by the author(s).

## 1. Introduction

Transformers (Vaswani et al., 2017) are the foundation of recent advances in machine learning across various domains (Jumper et al., 2021; Dosovitskiy et al., 2021; Comanici et al., 2025). This success often is attributed to their ability to learn at scale (Kaplan et al., 2020) and in-context (Brown et al., 2020), both of which are the byproduct of their primary building block–attention module–that acts as an associative memory with growing capacity (Ramsauer et al., 2021; Bietti et al., 2024; Behrouz et al., 2025b). While effective for many retrieval tasks (Arora et al., 2024b), this growing memory incurs quadratic complexity and high inference-time memory usage (KV-caching). This has motivated the development of sub-quadratic architectures that aim to improve efficiency while maintaining performance (Dai et al., 2019; Child et al., 2019; Poli et al., 2023).

In particular, recurrent neural networks that aim to compress the past data into their memory state, maintaining a fixed size over the entire input sequence, have regained attention in recent years (Katharopoulos et al., 2020; Irie et al., 2021; Sun et al., 2023; Behrouz et al., 2024). Despite showing promising preliminary results in diverse short-context language modeling and other sequence modeling tasks (Irie et al., 2022; Dalal et al., 2025), the fixed-memory state of such recurrent architectures is the bottleneck to unleash their actual power. The foundation of these architectures is based on recurrence and data compression, which, with careful design, can result in highly efficient and expressive learning algorithms (Merrill et al., 2024; Huang et al., 2024). However, their fixed capacity to compress a growing sequence forces them to forget past information, which is a critical bottleneck, specifically in recall-intensive and long-context tasks (Arora et al., 2024b; Kuratov et al., 2024; Sun et al., 2024).

**Contributions.** We introduce Memory Caching (MC), a general technique that allows the effective memory of recurrent models to grow with sequence length by caching checkpoints of the memory states (see Figure 1). MC provides a flexible middle ground interpolating between standard recurrence and attention, offering a controllable complexity of $\mathcal{O}(NL)$. This allows for flexible interpolation between the $\mathcal{O}(L)$ complexity of RNNs and the $\mathcal{O}(L^2)$ complexity of Transformers. Our contributions are threefold:

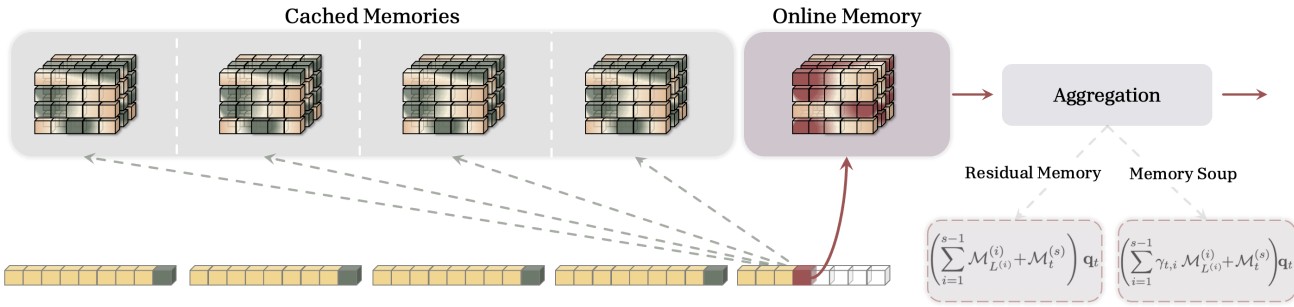

*Figure 1.* The Overall Memory Caching Method. Each token attends to its online memory as well as a set of cached memories from the past.

- **The MC Framework:** We propose segmenting the sequence and caching the compressed memory state of each segment, allowing the model to directly access compressed information from the entire history.

- **Novel Aggregation Strategies:** We introduce four methods to utilize these cached memories: (i, ii) (Gated) Residual Memory, which uses residual connections and a novel context-aware gating mechanism; (iii) Memory Soup, inspired by weight souping, which averages the parameters of cached memory modules (distinct for non-linear memories); and (iv) Sparse Selective Caching (SSC), which uses a Mixture-of-Experts style router to select only the most contextually relevant cached memories for efficient aggregation.

- **Empirical Validation:** As a proof of concept, we demonstrate the effectiveness of MC on three architectures of Linear Attention (LA) (Katharopoulos et al., 2020), deep memory module Titans (Behrouz et al., 2024) and Sliding Window Linear Attention (SWLA), and Deep Linear Attention (DLA) (Behrouz et al., 2025a), across language modeling, long-context, and retrieval tasks, showing that MC enhances performance and extends the effective context length of RNNs.

## 2. Preliminaries and Background

In this section, we review necessary background and establish notations.

**Notations.** We use bold lowercase (resp. uppercase) letters for vectors (resp. matrices) and use subscript $t$ to refer to the state of the entities correspond to time $t$. Throughout, we let $x \in \mathbb{R}^{L \times d_{in}}$ be the input, $\mathcal{M}_t$ be the state of memory $\mathcal{M}(\cdot)$ at time $t$, $\mathbf{K}$ be the keys, $\mathbf{V}$ be the values, $\mathbf{Q}$ be the query matrices, and $L$ denote the sequence length. We focus on MLP-based architectures for memory with $\mathcal{L}_{\mathcal{M}} \geq 1$ layers. Notably, this formulation includes linear matrix-valued memory modules when $\mathcal{L}_{\mathcal{M}} = 1$. When it is needed, we parameterize the memory module $\mathcal{M}(\cdot)$ with

$\boldsymbol{\theta}_{\mathcal{M}} := \{W_1, \dots, W_{\mathcal{L}_{\mathcal{M}}}, \dots\}$, which at least includes the parameters of linear layers in the MLP.

**Attention.** Attention (Vaswani et al., 2017) is the primary building block of Transformers that acts as their associative memory (Bietti et al., 2023; Behrouz et al., 2025b; Wang et al., 2025). Given input $x \in \mathbb{R}^{L \times d_{in}}$, causal attention computes output $\mathbf{y} \in \mathbb{R}^{L \times d_{in}}$ over input dependent key, value, and query matrices $\mathbf{Q} = x\mathbf{W_Q}, \mathbf{K} = x\mathbf{W_K}$, and $\mathbf{V} = x\mathbf{W_V}$ as:

$$\mathbf{y}_i = \sum_{t=1}^{i} \frac{\exp\left(\mathbf{q}_i^\top \mathbf{k}_t\right) \mathbf{v}_t}{\sum_{\ell=1}^{i} \exp\left(\mathbf{q}_i^\top \mathbf{k}_\ell\right)} = \frac{1}{Z_i} \sum_{t=1}^{i} \exp\left(\mathbf{q}_i^\top \mathbf{k}_t\right) \mathbf{v}_t, \tag{1}$$

where $\mathbf{W_Q}, \mathbf{W_K}$, and $\mathbf{W_V} \in \mathbb{R}^{d_{in} \times d_{in}}$ are learnable parameters, and $Z_i = \sum_{\ell=1}^{i} \exp\left(\mathbf{q}_i^\top \mathbf{k}_\ell\right)$ is the normalization term. Attention requires $\mathcal{O}(L^2)$ operations due to the need to access all past tokens.

**Linear Attention.** Linear attention (Katharopoulos et al., 2020) and its variants (Schlag et al., 2021; Peng et al., 2023; Yang et al., 2024b) improves efficiency of attention by replacing the $\exp(\cdot)$ operator in Equation 1 with a separable kernel $\phi(\cdot)$, resulting in an efficient recurrent formulation:

$$\mathbf{y}_i = \sum_{t=1}^{i} \frac{\phi\left(\mathbf{q}_i\right)^\top \phi\left(\mathbf{k}_t\right) \mathbf{v}_t}{\sum_{\ell=1}^{i} \phi\left(\mathbf{q}_i\right)^\top \phi\left(\mathbf{k}_\ell\right)} = \frac{1}{Z_i} \mathcal{M}_i \phi\left(\mathbf{q}_i\right), \tag{2}$$

where $\mathcal{M}_t = \mathcal{M}_{t-1} + \mathbf{v}_t \phi\left(\mathbf{k}_t\right)^\top$ acts as the fixed-size memory (Katharopoulos et al., 2020).

**Test-time Memorization Perspective.** A recent unifying framework interprets the forward pass of sequence models—including both attention and modern RNNs—as a dynamic learning/memorization process occurring at test time (Wang et al., 2025; Behrouz et al., 2025b). In this view, the model acts as an associative memory that actively learns the mapping between input tokens (keys and values). This memorization is achieved by optimizing an internal objective, often formalized as an $L_2$ regression problem (Sun

et al., 2024; Wang et al., 2025) or with a general objective of "attentional bias" (Behrouz et al., 2025b). This perspective frames the memory state as a dynamic entity optimized during the forward pass. Particularly, in the simplest form of Miras framework (Behrouz et al., 2025b), associative memory $\mathcal{M}(\cdot)$ aims to learn a mapping between keys $\{\boldsymbol{k}_t\}_{t=1}^{L}$ and values $\{\boldsymbol{v}_t\}_{t=1}^{L}$ based on an objective (called "attentional bias"):

$$\mathcal{M}_{t+1} = \arg\min_{\mathcal{M}} \ \mathcal{L}\left(\mathcal{M}(\boldsymbol{k}_t); \boldsymbol{v}_t\right) + \text{Ret}\left(\mathcal{M}; \mathcal{M}_t\right), \quad (3)$$

where objective $\mathcal{L}(\cdot)$ measures the quality of mapping and $\text{Ret}\left(\mathcal{M}; \mathcal{M}_t\right)$ keeps the new solution close to the last state of the memory. For particular choices of attentional bias, one can recover well-known architectures: For example, letting $\mathcal{L}\left(\mathcal{M}(\boldsymbol{k}_t); \boldsymbol{v}_t\right) = \langle \mathcal{M}(\boldsymbol{k}_t), \boldsymbol{v}_t \rangle$ and $\mathcal{M}(\cdot) \in \mathbb{R}^{d \times d}$, recovers unnormalized linear attention architecture (Katharopoulos et al., 2020). We leverage this view by introducing Memory Caching, where cached states serve as checkpoints of this optimization process, enhancing the model's ability to retrieve information across long sequences.

# 3. Recurrent Neural Networks with Memory Caching

RNNs maintain a fixed-size memory to compress the input sequence. As sequences grow long, this leads to memory overflow and performance degradation. Conversely, attention caches all past tokens, resulting in a growing memory but quadratic cost. We propose Memory Caching (MC) to cache intermediate memory states, providing a middle ground where the model's memory can grow with arbitrary scale. This allows computational costs to interpolate between $\mathcal{O}(L)$ (similar to RNNs) and $\mathcal{O}(L^2)$ (similar to Transformers). To this end, given a sequence of tokens $x \in \mathbb{R}^{L \times d_{\text{in}}}$, we split the sequence into segments $S^{(1)}, \ldots, S^{(N)}$ with size $L^{(1)}, \ldots, L^{(N)}$ and use memories $\mathcal{M}^{(1)}, \ldots, \mathcal{M}^{(N)}$ to compress the segments. The memory update rule or the recurrence for memory corresponds to $s$-th segment is:

$$\boldsymbol{k}_t = x_t W_{\boldsymbol{k}}, \qquad \boldsymbol{v}_t = x_t W_{\boldsymbol{v}}, \qquad \boldsymbol{q}_t = x_t W_{\boldsymbol{q}},$$
$$\mathcal{M}_t^{(s)} = f\left(\mathcal{M}_{t-1}^{(s)}; \boldsymbol{k}_t, \boldsymbol{v}_t\right), \text{ where } 1 \le t \le L^{(s)}, \quad (4)$$

where $f(\cdot)$ is the learning update rule (e.g., $f\left(\mathcal{M}_{t-1}^{(s)}; \boldsymbol{k}_t, \boldsymbol{v}_t\right) = \mathcal{M}_{t-1}^{(s)} + \boldsymbol{v}_t \boldsymbol{k}_t^{\top}$ for linear attention (Katharopoulos et al., 2020)). Using the above formulation, after updating the memories, we cache their last state in each segment (i.e., $\{\mathcal{M}_{L^{(s)}}^{(s)}\}_{s=1}^{T}$ where $T$ is the index of the current segment, $x_t \in S^{(T)}$). Standard RNNs compute the output using only the current memory state: $\mathbf{y}_t = \mathcal{M}_t(\boldsymbol{q}_t)$. In contrast, our formulation uses all cached memories alongside the current memory (online memory) to compute the output for query $\boldsymbol{q}_t$. Given an arbitrary aggregation function, $\text{Agg}(\cdot; \cdot; \cdot)$, the output is:

$$\mathbf{y}_t = \text{Agg}\left(\{\mathcal{M}_{L^{(1)}}^{(1)}(\cdot), \ldots, \mathcal{M}_{L^{(s-1)}}^{(s-1)}(\cdot)\}; \mathcal{M}_t^{(s)}(\cdot); \mathbf{q}_t\right),$$

where $s$ is the indices of the current segment. Note that for $1 \le i \le s$, the term $\mathcal{M}_{L^{(i)}}^{(i)}(\mathbf{q}_t)$ provides us with the corresponding information to query $\boldsymbol{q}_t$ in segment $i$. In the following sections, we present different effective choices of $\text{Agg}(\cdot; \cdot; \cdot)$ function to incorporate the past information into the computation of the current output and increasing the effective memory capacity of the model.

## 3.1. Residual Memory

We begin with the simplest $\text{Agg}(\cdot; \cdot; \cdot)$ operator: a summation, acting as a residual connection across memory states. In this case, given keys, values, and queries (see Equation 4) and segments $S^{(1)}, \ldots, S^{(N)}$, we define the memory update and output computation at time $t$ in segment $s$ as:

$$\mathcal{M}_t^{(s)} = f\left(\mathcal{M}_{t-1}^{(s)}; \boldsymbol{k}_t, \boldsymbol{v}_t\right), \qquad \text{where} \quad 1 \le t \le L^{(s)},$$

$$\mathbf{y}_t = \underbrace{\mathcal{M}_t^{(s)}(\mathbf{q}_t)}_{\text{Online Memory}} + \underbrace{\sum_{i=1}^{s-1} \mathcal{M}_{L^{(i)}}^{(i)}(\mathbf{q}_t)}_{\text{Cached Memories}}. \quad (5)$$

The critical change in memory caching is how the output is computed. In fact, for retrieval of the memory, the model uses forward passes over both the current memory (called online memory) and the cached memories for input query $\boldsymbol{q}_t$.

**Gated Residual Memory (GRM).** When the memory module is strictly linear (i.e., $\mathcal{M}$ is a matrix), the Residual Memory formulation (Equation 5) mathematically collapses into a standard fixed-size memory, as the cached memories can be pre-summed (c.f. Equation 11 below). However, in practice, our experimental results show that even this simple formulation can enhance the power of recurrent models (see Section 4). The main reason is even a simple residual memory acts as a retention operator that enhance access to the long past. A further limitation of the residual approach is that it treats all cached memories equally, ignoring their relevance to the query $\boldsymbol{q}_t$. To enable selective retrieval, we introduce input-dependent gating. Given input $x_t$ in segment $s$, we define parameters $0 \le \gamma_t^{(1)}, \ldots, \gamma_t^{(s)} \le 1$ be input-dependent parameters and reformulate the output as:

$$\mathcal{M}_t^{(s)} = f\left(\mathcal{M}_{t-1}^{(s)}; \boldsymbol{k}_t, \boldsymbol{v}_t\right), \text{ for } 1 \le t \le L^{(s)}, \quad (6)$$

$$\mathbf{y}_t = \gamma_t^{(s)} \mathcal{M}_t^{(s)}(\mathbf{q}_t) + \sum_{i=1}^{s-1} \gamma_t^{(i)} \mathcal{M}_{L^{(i)}}^{(i)}(\mathbf{q}_t) \quad (7)$$

Here, parameters $\gamma_t^{(i)}$ modulate the contribution of each segment to the output. When $\gamma_t^{(i)} \to 1$ (resp. $\gamma_t^{(i)} \to 0$),

$i$-th segment has more (resp. less) contribution to the output. Due to these input dependent parameters, the above formulation cannot be pre-computed before this token and also cannot be reused for next tokens/segments. Therefore, contrary to the previous variant, it does not collapse into the fixed-size memory case (even in the linear memory case) and so requires to be recomputed for every token and needs caching memory states. A simple choice of parametrization for $\gamma_t^{(i)}$s is to define them as linear projection of input $x_t$ (similar to projections for keys, values, and queries). With this parametrization, however, $\gamma_t^{(i)}$ acts as a position-based filtering/focus, meaning that the context of $x_t$ only determines how much the $i$-th segment's memory (based on the position) contributes, no matter what its context is. To overcome this issue, we suggests making $\gamma_t^{(i)}$ as a function of both $x_t$ and $i$-th segment $S^{(i)}$, incorporating both of their contexts and how similar they are. To this end, we introduce a connector parameter $\boldsymbol{u}_t$ as the linear projection of input, and define $\gamma_t^{(i)}$ as the similarity of $\boldsymbol{u}_t$ and $i$-th segment $S^{(i)}$:

$$\gamma_t^{(i)} = \langle \boldsymbol{u}_t, \texttt{MeanPooling}(S^{(i)}) \rangle \text{ where } \boldsymbol{u}_t = x_t W_{\boldsymbol{u}} \, . \tag{8}$$

Here, $\texttt{MeanPooling}(\cdot)$ provides a simple representation of segment's context as the mean of all tokens. It, however, can be replaced by any other pooling process. In practice, we also normalize $\gamma_t^{(i)}$ using $\texttt{softmax}(\cdot)$. As an alternative parametrization, we can use $\boldsymbol{u}_t = \boldsymbol{q}_t$. When $\gamma_t^{(i)}$s are constant, then GRM is equivalent to residual memory.

**Example.** To better illustrate the above formulations and as an illustrative example, we let $f\left(\mathcal{M}_{t-1}^{(s)}; \boldsymbol{k}_t, \boldsymbol{v}_t\right) = \mathcal{M}_{t-1}^{(s)} - \nabla\langle\mathcal{M}_{t-1}^{(s)}(\boldsymbol{k}_t), \boldsymbol{v}_t\rangle$, where memory $\mathcal{M}(\cdot)$ is an arbitrary feedforward layer (e.g., MLP or gated MLP layers). This general form is equivalent to Deep Linear Attention (DLA) (Behrouz et al., 2025a) and when the memory is a matrix (i.e., MLP with one layer) it is equivalent to the linear attention (Katharopoulos et al., 2020). Using residual memory caching on DLA results in a model with update and retrieval rules of:

$$\mathcal{M}_t^{(s)} = \mathcal{M}_{t-1}^{(s)} - \nabla\langle\mathcal{M}_{t-1}^{(s)}(\boldsymbol{k}_t), \boldsymbol{v}_t\rangle,$$
$$\mathbf{y}_t = \mathcal{M}_t^{(s)}(\mathbf{q}_t) + \sum_{i=1}^{s-1} \mathcal{M}_{L^{(i)}}^{(i)}(\mathbf{q}_t) \, . \tag{9}$$

When using a linear matrix-valued memory (i.e., linear attention), Equation 9 can be simplified to:

$$\mathcal{M}_t^{(s)} = \mathcal{M}_{t-1}^{(s)} + \boldsymbol{v}_t \boldsymbol{k}_t^\top, \tag{10}$$

$$\mathbf{y}_t = \mathcal{M}_t^{(s)}\mathbf{q}_t + \sum_{i=1}^{s-1} \mathcal{M}_{L^{(i)}}^{(i)}\mathbf{q}_t = \left(\mathcal{M}_t^{(s)} + \sum_{i=1}^{s-1} \mathcal{M}_{L^{(i)}}^{(i)}\right)\mathbf{q}_t \, . \tag{11}$$

**Memory Complexity.** In the retrieval process, we use the current memory (online memory) and the cached memories of all previous segments and so given a fixed training sequence length, the number of cached memories is a function of segment lengths. While the memory update process has not changed and so requires $\mathcal{O}(L)$ operation, the retrieval process requires forward pass over all cached memory and so needs $\mathcal{O}(N)$ operations per token. This brings the complexity of the model to $\mathcal{O}(NL)$, where $1 \le N \le L$. Note that when $N = 1$ (only one segment), we do not need to cache any memory state, resulting in a simple recurrent memory model. When $N = L$, it means that each token is treated as a separate segment and so the memory state for all past tokens are cached. This closely matches the intuition behind the power of attention. In fact, attention by caching all past tokens, provide a direct access to each part of the sequence, enhancing the retrieval ability.

### 3.2. Memory Soup

Viewing the recurrence as a meta-learning process where memory states are checkpoints, we introduce the Memory Soup variant, inspired by Wortsman et al. (2022). The core idea is to combine the memory states (parameters) into a single data-dependent memory for retrieval. Similar to the previous variant, we use $\mathcal{M}_{L^{(i)}}^{(i)}$ to refer to the cached memory corresponds to $i$-th segment and parameterize it with $\boldsymbol{\theta}_{\mathcal{M}_{L^{(i)}}^{(i)}} := \{W_1^{(i)}, \ldots, W_c^{(i)}\}$. Note that the architecture of memory is unchanged and so $c$ (the number of parameters) is the same for all memory states. Accordingly, the memory update and retrieval process for MC is defined as:

$$\mathcal{M}_t^{(s)} = f\left(\mathcal{M}_{t-1}^{(s)}; \boldsymbol{k}_t, \boldsymbol{v}_t\right), \text{ for } 1 \le t \le L^{(s)}, \tag{12}$$
$$\mathbf{y}_t = \mathcal{M}_t^*(\boldsymbol{q}_t), \tag{13}$$

where $\mathcal{M}_t^*$ is parametrized as: $\boldsymbol{\theta}_{\mathcal{M}_t^*} := \left\{\sum_{i=1}^s \gamma_t^{(i)} W_1^{(i)}, \ldots, \sum_{i=1}^s \gamma_t^{(i)} W_c^{(i)}\right\}$. Therefore, each token has its own memory for retrieval that also depends on the input-data and can change. In fact, one can interpret the above process as *a memory system that each token, builds its own memory to retrieve corresponding information from*. Note that here $\gamma_t^{(i)}$ parameters are defined with the same process as Equation 8.

When the memory module $\mathcal{M}$ is linear, Memory Soup is mathematically equivalent to GRM (Equation 7). This is because souping the weights and then applying the query is identical to applying the query to individual memories and then ensembling the outputs, due to the linearity of the operation. The distinction becomes crucial when using deep or non-linear memory modules (e.g., DLA or Titans). In these cases, the equivalence breaks down. Memory Soup constructs a new, input-dependent memory module $\mathcal{M}_t^*$ by interpolating the parameters themselves, effectively creating

a specialized non-linear retrieval function tailored for that specific timestep.

### 3.3. Sparse Selective Caching (SSC) of Memories

The previous variants attend to all past cached memories, which can cause significant memory overhead for ultra-long sequences. We introduce Sparse Selective Caching (SSC), where each token contextually *selects* a subset of cached memories, improving efficiency. To this end, inspired by Mixture of Experts (MoEs) (Shazeer et al., 2017), we use a router that based on the token and its similarity to the context of each segment choose a subset of cached memories. For each segment $S^{(i)}$, following Equation 8, we let $\texttt{MeanPooling}(S^{(i)}) = \sum_{j \in S^{(i)}} \boldsymbol{k}_j$, and define the relevance score of each segment $S^{(i)}$ to query $x_t$ as:

$$\mathbf{r}_t^{(i)} = \langle \boldsymbol{u}_t, \texttt{MeanPooling}(S^{(i)}) \rangle, \quad \text{where} \quad \boldsymbol{u}_t = x_t W_{\boldsymbol{u}} .$$

Given relevance scores, the router chooses $k$ of the cached memories with highest relevance, i.e., $\mathcal{R}_t = \arg \texttt{Top-k}(\{\mathbf{r}_t^{(i)}\}_{i=1}^{s-1})$, as well as the current online memory for retrieval. Given selected memories, the retrieval process is the same as previous variants but using only selected memories:

$$\mathcal{M}_t^{(s)} = f\left(\mathcal{M}_{t-1}^{(s)}; \boldsymbol{k}_t, \boldsymbol{v}_t\right), \qquad \text{where} \quad 1 \le t \le L^{(s)} ,$$
$$\mathbf{y}_t = \gamma_t^{(s)} \mathcal{M}_t^{(s)}(\mathbf{q}_t) + \sum_{i \in \mathcal{R}_t} \gamma_t^{(i)} \mathcal{M}_{L^{(i)}}^{(i)}(\mathbf{q}_t) . \tag{14}$$

In this formulation, $\texttt{MeanPooling}(S^{(i)})$ of each segment can be pre-computed and so computing the relevance score as well as choosing $\texttt{Top-k}$ segments for each token are simply parallelizable. Also, such computations do not require to store the state of the cached memories in the accelerators (i.e., GPUs, TPUs, etc.). Therefore, this process only requires loading the *"selected"* memories for each token and so can enhance memory consumption during both training and inference.

**Effective Memory.** One interesting interpretation of SSC is to see it as a sparse unified memory module. We illustrate this in Figure 2 (Right). One can see SSC as a model with growing memory, where for each token activates a subset of parameters for memory write operation (storing the token), and a larger subset of parameters for retrieval. This formulation allows the memory to (1) store information without any interfering with past memories, and (2) efficiently and adaptively retrieve information. The segment size, here, determines the size of the blocks in the unified memory that become active together.

### 3.4. Checkpoints or Independent Compressors?

Revisiting the general formulation of the memory caching in Equation 4, there is a design choice that if we cache

the checkpoints of a single memory, or we use a set of independent memory modules to compress the information each segment. That is, there are two perspectives:

**(1)** From the optimization point of view (i.e., Equation 3), we are training an associative memory and the tokens in the sequence are the training data samples. Accordingly, to avoid forgetting past knowledge, we cache the checkpoints of the memory through training (optimization process). In this formulation, for each $s = 0, \cdots, N$, we have $\mathcal{M}_0^{(s)}(\cdot) = \mathcal{M}_{L^{(s-1)}}^{(s-1)}(\cdot)$, meaning that memory, in each segment, starts from its last state in the previous segment.

**(2)** From the compression perspective, when we cache a memory of a past segment, we want it to be a compressed representative of the information in that segment. Therefore, the forward pass $\mathcal{M}_{L^{(s)}}^{(s)}(\boldsymbol{q}_t)$ represents the corresponding information to $\boldsymbol{q}_t$ in $s$-th segment. To avoid interference of memories in different segments, we use independent memories for each segment, meaning that for segment $s$, the memory starts from an initial point $\mathcal{M}_0^{(s)}(\cdot)$ that is independent of $\mathcal{M}_{L^{(s-1)}}^{(s-1)}(\cdot)$.

In practice, we observe that each of these two choices has its own advantageous/disadvantageous (see Section 4.7).

### 3.5. Memory Caching for Titans and Linear Attention Variants

As discussed earlier in the paper, the memory caching technique is applicable to any arbitrary recurrent update rule and potentially can enhance their effective context length. As a proof of concept, we use use memory caching on the update rule of Sliding Window Linear Attention (SWLA) and Deep Linear Attention (DLA) (Behrouz et al., 2025a), Linear Attention (Katharopoulos et al., 2020), and Titans (Behrouz et al., 2024).

**Sliding Window Linear Attention.** Recently, Behrouz et al. (2025a) introduced Sliding Window Linear Attention (SWLA), in which the memory updates its weights based on a set of $c \ge 1$ past tokens (contrary to the online RNNs, where the memory is updated solely based on the last token). More specifically, given a memory module $\mathcal{M}(\cdot)$ and keys, values, and queries, $\{(\boldsymbol{k}_t, \boldsymbol{v}_t, \boldsymbol{q}_t)\}_{t=1}^{L}$, the update and retrieval rules are defined as:

$$\mathcal{M}_t = \alpha_t \mathcal{M}_{t-1} + \sum_{i=t-c+1}^{t} \beta_i^{(t)} \boldsymbol{v}_i \boldsymbol{k}_i^{\top}, \tag{15}$$
$$\mathbf{y}_t = \mathcal{M}_t \boldsymbol{q}_t, \tag{16}$$

When $c = 1$, the design collapses to the simple linear attention (online linear RNN) and its gated variants (Katharopoulos et al., 2020; Yang et al., 2024b; Li et al., 2025). As a proof of concept, we use memory caching on SWLA with

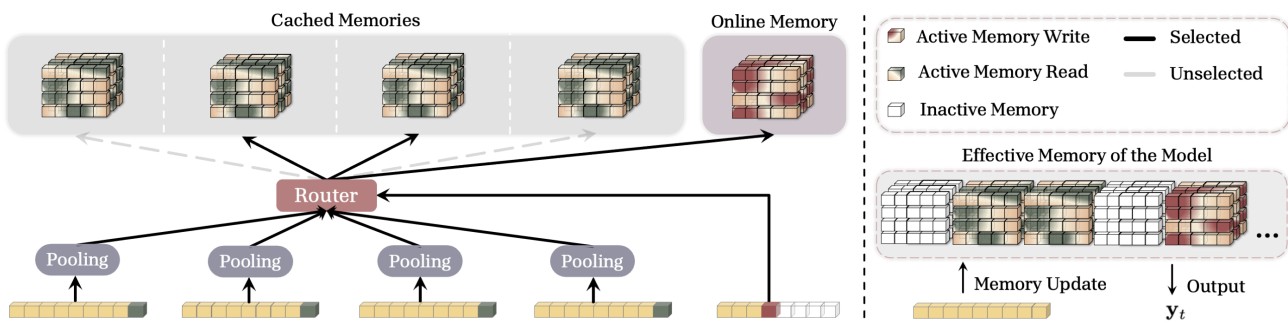

*Figure 2.* Sparse Selective Caching (SSC) of Memories. A router measures the contextual similarity of each token to its past segments and chooses a subset of past cached memory for better efficiency.

$c = 2$, resulting in a recurrence and retrieval of:

$$\mathcal{M}_t^{(s)} = \alpha_t \, \mathcal{M}_{t-1}^{(s)} + \left( \beta_t \boldsymbol{v}_{t-1} \boldsymbol{k}_{t-1}^\top + \lambda_t \boldsymbol{v}_t \boldsymbol{k}_t^\top \right), \quad (17)$$

$$\mathbf{y}_t = \gamma_t^{(s)} \mathcal{M}_t^{(s)} \boldsymbol{q}_t + \sum_{i=1}^{s-1} \gamma_t^{(i)} \mathcal{M}_{L^{(i)}}^{(i)} \boldsymbol{q}_t. \quad (18)$$

Note that, as discussed earlier, since SWLA is a linear memory module, both GRM and Memory Soup variants result in the same formulation.

**Deep Linear Attention.** DLA uses the same update rule as linear attention (i.e., Hebbian rule), but with a deep memory module. That is, given a memory module $\mathcal{M}(\cdot)$ and keys, values, and queries, $\{(\boldsymbol{k}_t, \boldsymbol{v}_t, \boldsymbol{q}_t)\}_{t=1}^L$, the update and retrieval rules are defined as:

$$\mathcal{M}_t = \mathcal{M}_{t-1} - \eta_t \nabla \mathcal{L} \left( \mathcal{M}_{t-1}; \boldsymbol{k}_t, \boldsymbol{v}_t \right), \quad (19)$$

$$\mathbf{y}_t = \mathcal{M}_t(\boldsymbol{q}_t), \quad (20)$$

where the attentional bias objective is defined as $\mathcal{L}(\mathcal{M}_{t-1}; \boldsymbol{k}_t, \boldsymbol{v}_t) = -\langle \mathcal{M}_{t-1}(\boldsymbol{k}_t), \boldsymbol{v}_t \rangle$. Using memory caching (GRM variant), the update and retrieval process for DLA are defined as:

$$\mathcal{M}_t^{(s)} = \mathcal{M}_{t-1}^{(s)} - \eta_t \nabla \mathcal{L} \left( \mathcal{M}_{t-1}^{(s)}; \boldsymbol{k}_t, \boldsymbol{v}_t \right), \text{ for } 1 \leq t \leq L^{(s)},$$
$$(21)$$

$$\mathbf{y}_t = \gamma_t^{(s)} \mathcal{M}_t^{(s)}(\mathbf{q}_t) + \sum_{i=1}^{s-1} \gamma_t^{(i)} \mathcal{M}_{L^{(i)}}^{(i)}(\mathbf{q}_t). \quad (22)$$

Similarly, we can replace Equation 12 or Equation 14 with the update rule of DLA (similar to Equation 40) to derive other memory caching variants. Note that when memory module $\mathcal{M}(\cdot)$ is a matrix, then the above formulation is equivalent to the linear attention (Katharopoulos et al., 2020).

**Titans.** In Titans, compared to DLA, both attentional bias objective as well as the internal optimizer are different. More specifically, given a memory module $\mathcal{M}(\cdot)$ and keys,

values, and queries, $\{(\boldsymbol{k}_t, \boldsymbol{v}_t, \boldsymbol{q}_t)\}_{t=1}^L$, the update and retrieval rules of Titans are defined as:

$$\mathcal{M}_t = \alpha_t \, \mathcal{M}_{t-1} - \mathcal{S}_t, \quad (23)$$

$$\mathcal{S}_t = \beta_t \, \mathcal{S}_{t-1} - \eta_t \nabla \mathcal{L} \left( \mathcal{M}_{t-1}; \boldsymbol{k}_t, \boldsymbol{v}_t \right), \quad (24)$$

$$\mathbf{y}_t = \mathcal{M}_t(\boldsymbol{q}_t), \quad (25)$$

where the attentional bias objective is defined as $\mathcal{L}(\mathcal{M}_{t-1}; \boldsymbol{k}_t, \boldsymbol{v}_t) = \|\mathcal{M}_{t-1}(\boldsymbol{k}_t) - \boldsymbol{v}_t\|_2^2$. With MC, while the memory update operation for each segment is the same as Equation 42 and Equation 43, the retrieval process for Titans with memory caching is defined the same as Equation 41.

**Log-Linear++ Variant.** Recently, Guo et al. (2025) take advantage of structured matrix formulation of linear RNNs and design log-linear attention, a hierarchical algorithm based on Fenwick tree structure (Fenwick, 1994) that allows a logarithmically growing set of hidden states. We aim to use log-linear attention as the baseline in our experiments to show the effect of segmentation on the efficiency and retrieval performance. Its formulation, however, suffer from the positional bias and lack of context-dependency in retrieval process that we discussed in Section 3.1. For the sake of fair comparison, we improve Log-linear formulation, denoted as Log-Linear++ in our experiments, by reformulating it as a variant of memory caching with GRM and logartithmic-size set of segments. The segmentation process is the same as the process describe in Section A.2. The other components are kept unchanged, the same as our memory caching variants.

## 4. Experiments

Next, we evaluate the effectiveness of memory caching in improving the performance of models on language modeling, commonsense reasoning, needle in haystack, and in-context recall tasks.

**Experimental Setup.** We train our models with training context window of size {2K, 4K, 8K, 16K, 32K} and seg-

*Table 1.* Performance of models on language modeling and common-sense reasoning tasks.

| Model | Wiki. ppl ↓ | LMB. ppl ↓ | LMB. acc ↑ | PIQA acc ↑ | Hella. acc_n ↑ | Wino. acc ↑ | ARC-e acc ↑ | ARC-c acc_n ↑ | SIQA acc ↑ | BoolQ acc ↑ | Avg. ↑ |
|---|---|---|---|---|---|---|---|---|---|---|---|
| | | | | | 760M params / 30B tokens | | | | | | |
| Transformer++ | 24.18 | 24.27 | 36.3 | 67.2 | 41.8 | 52.0 | 65.6 | 33.4 | 39.1 | 61.7 | 49.64 |
| RetNet | 25.77 | 24.19 | 34.5 | 66.8 | 41.2 | 51.9 | 63.6 | 32.5 | 38.8 | 56.2 | 48.19 |
| DeltaNet | 24.52 | 24.38 | 36.8 | 67.3 | 44.5 | 51.8 | 64.2 | 32.7 | 39.6 | 60.1 | 49.63 |
| RWKV-7 | 23.75 | 23.08 | 37.1 | 67.3 | 47.6 | 52.2 | 64.7 | 34.2 | 39.4 | 61.9 | 50.55 |
| Miras (Memora) | 22.28 | 22.31 | 38.2 | 67.8 | 49.3 | 53.3 | 63.6 | 36.1 | 40.9 | 63.0 | 51.53 |
| Samba* | 21.07 | 22.85 | 39.2 | 68.9 | 47.8 | 53.1 | 65.8 | 34.9 | 38.9 | 63.1 | 51.46 |
| SWLA | 23.83 | 22.74 | 36.5 | 66.9 | 44.1 | 54.9 | 64.2 | 34.1 | 39.6 | 60.1 | 50.05 |
| + Log-Linear++ | 23.37 | 22.19 | 36.9 | 67.3 | 44.7 | 55.0 | 64.9 | 34.6 | 39.4 | 60.4 | 50.40 |
| + GRM (= Soup) | 22.81 | 21.50 | 37.8 | 68.3 | 45.8 | 55.0 | 65.4 | 36.2 | 40.6 | 61.0 | 51.26 |
| + SSC | 23.06 | 22.39 | 37.2 | 67.9 | 45.2 | 54.9 | 65.2 | 35.5 | 39.8 | 60.6 | 50.79 |
| DLA | 23.12 | 22.09 | 36.1 | 68.0 | 47.9 | 52.7 | 65.8 | 34.6 | 39.1 | 59.6 | 50.48 |
| + Log-Linear++ | 23.08 | 21.15 | 36.8 | 68.1 | 47.7 | 53.0 | 65.6 | 35.1 | 39.2 | 59.3 | 50.60 |
| + GRM | 22.91 | 20.10 | 37.5 | 69.2 | 48.7 | 52.8 | 66.1 | 36.8 | 40.3 | 59.9 | 51.41 |
| + Memory Soup | 22.78 | 20.49 | 37.2 | 69.6 | 48.3 | 53.4 | 65.8 | 36.5 | 39.6 | 60.2 | 51.33 |
| + SSC | 23.14 | 20.86 | 37.0 | 68.4 | 47.7 | 52.7 | 66.0 | 35.2 | 39.7 | 60.1 | 50.85 |
| Titans (LMM) | 20.04 | 21.96 | 37.4 | 69.3 | 48.5 | 52.3 | 66.3 | 35.8 | 40.1 | 62.8 | 51.56 |
| + Log-Linear++ | 19.79 | 20.62 | 37.8 | 70.1 | 48.0 | 52.5 | 66.8 | 35.6 | 40.3 | 62.8 | 51.74 |
| + GRM | 19.14 | 20.21 | 38.3 | 70.6 | 48.4 | 54.0 | 67.5 | 36.4 | 41.7 | 63.5 | 52.55 |
| + Memory Soup | 19.52 | 20.38 | 38.0 | 71.4 | 48.6 | 53.7 | 67.1 | 35.4 | 41.3 | 63.1 | 52.33 |
| + SSC | 19.39 | 20.46 | 37.7 | 70.9 | 48.7 | 53.5 | 66.9 | 36.3 | 41.2 | 63.1 | 52.29 |
| | | | | | 1.3B params / 100B tokens | | | | | | |
| Transformer++ | 17.92 | 17.73 | 42.6 | 71.4 | 51.3 | 54.1 | 69.9 | 36.0 | 41.8 | 58.4 | 53.19 |
| RetNet | 18.91 | 17.04 | 41.2 | 71.3 | 49.1 | 55.2 | 67.5 | 34.1 | 41.4 | 61.0 | 52.60 |
| DeltaNet | 18.62 | 17.10 | 41.6 | 70.1 | 49.4 | 52.7 | 67.6 | 35.2 | 39.7 | 54.8 | 51.39 |
| Miras (Memora) | 15.90 | 12.04 | 48.7 | 73.1 | 56.0 | 57.4 | 71.5 | 37.9 | 40.2 | 61.3 | 55.76 |
| Samba* | 16.15 | 13.21 | 45.2 | 71.5 | 53.8 | 55.8 | 69.1 | 36.7 | 40.6 | 63.0 | 54.46 |
| SWLA | 18.47 | 16.23 | 39.4 | 70.9 | 48.8 | 56.5 | 67.3 | 35.8 | 41.5 | 60.2 | 52.55 |
| + Log-Linear++ | 18.67 | 16.09 | 39.9 | 71.2 | 49.3 | 56.6 | 68.1 | 36.3 | 41.4 | 60.4 | 52.90 |
| + GRM (= Soup) | 18.51 | 15.95 | 40.6 | 72.6 | 50.5 | 57.8 | 69.5 | 40.8 | 42.8 | 62.2 | 54.60 |
| + SSC | 18.61 | 16.01 | 40.4 | 71.9 | 50.0 | 57.1 | 68.9 | 38.6 | 42.2 | 61.2 | 53.79 |
| DLA | 16.31 | 12.29 | 44.5 | 70.6 | 53.9 | 54.2 | 69.6 | 36.0 | 40.8 | 60.2 | 53.72 |
| + Log-Linear++ | 16.22 | 12.25 | 44.9 | 71.1 | 54.5 | 54.8 | 70.0 | 36.6 | 41.3 | 60.7 | 54.24 |
| + GRM | 16.08 | 12.10 | 45.8 | 72.5 | 55.9 | 55.8 | 71.5 | 41.2 | 42.8 | 62.2 | 55.96 |
| + Memory Soup | 16.16 | 12.17 | 45.6 | 71.9 | 55.4 | 55.6 | 70.9 | 37.7 | 42.0 | 61.5 | 55.08 |
| + SSC | 16.20 | 12.19 | 45.3 | 71.7 | 54.8 | 55.3 | 70.4 | 37.1 | 41.4 | 61.1 | 54.64 |
| Titans (LMM) | 15.60 | 11.41 | 49.1 | 73.1 | 56.3 | 59.8 | 72.4 | 40.8 | 42.1 | 61.0 | 56.82 |
| + Log-Linear++ | 15.49 | 11.38 | 49.4 | 73.6 | 56.5 | 60.3 | 72.8 | 41.1 | 42.5 | 61.3 | 57.19 |
| + GRM | 15.37 | 11.29 | 50.4 | 74.5 | 57.4 | 61.5 | 73.8 | 42.6 | 43.9 | 62.5 | 58.33 |
| + Memory Soup | 15.42 | 11.31 | 49.9 | 74.2 | 57.3 | 60.8 | 73.5 | 42.2 | 43.4 | 62.0 | 57.91 |
| + SSC | 15.44 | 11.35 | 49.6 | 73.8 | 57.0 | 60.6 | 73.1 | 41.9 | 42.8 | 61.8 | 57.58 |

\* is a hybrid of attention + linear RNN (Ren et al., 2024).

ment lengths ranging from {16, 32, 64, 128, 256, 512} tokens using a mixture of FineWeb dataset (Penedo et al., 2024) and Long-Data-Collections (Together AI, 2024). In language modeling and common-sense reasoning tasks (Table 1), the default model is trained with context length of 4K and segment length of 256. We use model size of 760M, and 1.3B parameters and train them on 30B and 100B tokens sampled from the FineWeb dataset (Penedo et al., 2024). Perplexity is measured on held-out validation

data. As for the downstream tasks, we evaluate trained models on Wikitext (Merity et al., 2017), LMB (Paperno et al., 2016), PIQA (Bisk et al., 2020), HellaSwag (Zellers et al., 2019), WinoGrande (Sakaguchi et al., 2021), ARC-easy (ARC-e) and ARC-challenge (ARC-c) (Clark et al., 2018), SIQA (Sap et al., 2019), and BoolQ (Clark et al., 2019). In other downstream tasks such as Needle-in-a-haystack, in-context retrieval, and LongBench, we train the models with 16K context length to better distinguish the performance of

*Table 2.* Needle-In-A-Haystack experiments with three levels of difficulty: single-needle tasks—S-NIAH-1 (passkey retrieval), S-NIAH-2 (numerical needle), and S-NIAH-3 (UUID-based needle).

| Model | S-NIAH-1 (pass-key retrieval) | | | S-NIAH-2 (number in haystack) | | | S-NIAH-3 (uuid in haystack) | | |
|---|---|---|---|---|---|---|---|---|---|
| | 4K | 8K | 16K | 4K | 8K | 16K | 4K | 8K | 16K |
| Transformer | 88.6 | 76.4 | 79.8 | 100 | 98.8 | 94.2 | 78.0 | 69.2 | 40.8 |
| DLA | 96.4 | 71.2 | 44.0 | 79.6 | 42.6 | 28.2 | 18.2 | 8.8 | 4.0 |
| + Log-Linear++ | 100 | 96.2 | 70.4 | 87.6 | 70.4 | 18.0 | 28.8 | 20.4 | 6.0 |
| + GRM | 100 | 100 | 82.4 | 94.6 | 82.8 | 54.8 | 48.2 | 34.4 | 18.2 |
| + Memory Soup | 100 | 100 | 78.2 | 91.8 | 77.2 | 40.4 | 43.0 | 32.8 | 14.8 |
| + SSC | 100 | 98.2 | 76.8 | 89.2 | 74.8 | 37.6 | 34.0 | 28.6 | 11.2 |
| Titans (LMM) | 100 | 100 | 100 | 99.6 | 84.6 | 75.4 | 74.2 | 42.8 | 21.2 |
| + Log-Linear++ | 100 | 100 | 100 | 95.6 | 88.4 | 74.8 | 76.0 | 48.4 | 24.2 |
| + GRM | 100 | 100 | 100 | 99.8 | 96.6 | 88.2 | 89.4 | 69.0 | 32.2 |
| + Memory Soup | 100 | 100 | 100 | 98.8 | 92.2 | 83.0 | 84.2 | 61.8 | 28.6 |
| + SSC | 100 | 100 | 100 | 98.6 | 90.4 | 79.6 | 81.0 | 54.2 | 27.0 |

*Table 3.* Accuracy on retrieval tasks w/ input truncated to different lengths.

| Model | SWDE | | | | SQuAD | | | | FDA | | | |
|---|---|---|---|---|---|---|---|---|---|---|---|---|
| | 512 | 1024 | 2048 | 16k | 512 | 1024 | 2048 | 16k | 512 | 1024 | 2048 | 16k |
| Transformer | 46.2 | 43.7 | 44.4 | 44.0 | 33.1 | 33.3 | 33.6 | 33.4 | 71.0 | 69.5 | 71.6 | 71.0 |
| Titans (MAL) | 51.9 | 48.6 | 48.3 | 48.5 | 28.3 | 29.2 | 29.1 | 28.8 | 71.1 | 73.9 | 72.1 | 71.7 |
| DLA | 44.5 | 39.9 | 32.7 | 32.5 | 23.8 | 24.0 | 23.8 | 24.1 | 55.6 | 40.2 | 25.9 | 23.3 |
| + Log-Linear++ | 43.7 | 37.7 | 30.4 | 30.6 | 27.8 | 27.8 | 27.9 | 28.3 | 55.4 | 39.6 | 22.3 | 18.9 |
| + GRM | 52.4 | 48.9 | 48.7 | 48.5 | 29.5 | 30.7 | 30.7 | 30.1 | 63.3 | 51.6 | 48.9 | 41.5 |
| + Memory Soup | 49.5 | 45.0 | 38.0 | 37.7 | 28.4 | 28.6 | 28.5 | 29.1 | 60.5 | 48.4 | 37.2 | 34.6 |
| + SSC | 47.0 | 42.5 | 35.5 | 35.3 | 26.0 | 28.1 | 27.1 | 28.8 | 58.0 | 46.0 | 28.8 | 29.4 |
| Titans (LMM) | 43.2 | 34.4 | 29.2 | 29.7 | 25.7 | 26.2 | 26.3 | 25.6 | 59.3 | 45.5 | 35.4 | 32.5 |
| + Log-Linear++ | 48.0 | 41.4 | 37.2 | 37.0 | 27.2 | 27.3 | 27.2 | 27.1 | 67.0 | 55.5 | 41.2 | 32.4 |
| + GRM | 52.6 | 49.3 | 49.5 | 50.1 | 29.7 | 30.4 | 31.5 | 32.0 | 72.9 | 68.7 | 61.1 | 52.6 |
| + Memory Soup | 50.3 | 46.7 | 44.8 | 45.4 | 29.2 | 29.7 | 29.8 | 30.3 | 70.3 | 63.8 | 55.7 | 45.8 |
| + SSC | 48.6 | 44.2 | 41.0 | 41.4 | 28.3 | 28.8 | 28.5 | 28.8 | 68.2 | 59.4 | 47.6 | 38.9 |

| Model | TriviaQA | | | | Drop | | | | NQ | | | Avg. |
|---|---|---|---|---|---|---|---|---|---|---|---|---|
| | 512 | 1024 | 2048 | 16k | 512 | 1024 | 2048 | 16k | 512 | 1024 | 2048 | |
| Transformer | 47.5 | 48.5 | 47.4 | 47.6 | 21.8 | 22.0 | 21.5 | 21.4 | 23.6 | 23.1 | 23.7 | 41.00 |
| Titans (MAL) | 44.8 | 45.1 | 44.6 | 44.8 | 20.6 | 20.5 | 20.8 | 20.9 | 22.1 | 22.4 | 22.5 | 40.46 |
| DLA | 43.3 | 44.2 | 43.5 | 43.2 | 20.1 | 19.9 | 20.6 | 20.0 | 19.7 | 18.4 | 18.5 | 30.51 |
| + Log-Linear++ | 43.7 | 44.8 | 43.6 | 43.8 | 20.3 | 20.2 | 20.8 | 20.2 | 19.9 | 18.8 | 21.0 | 30.75 |
| + GRM | 50.1 | 47.3 | 44.8 | 50.0 | 21.9 | 21.8 | 22.0 | 21.7 | 23.5 | 23.3 | 23.4 | 38.03 |
| + Memory Soup | 48.0 | 46.4 | 44.2 | 48.7 | 21.5 | 21.3 | 21.7 | 21.2 | 22.8 | 22.4 | 22.5 | 35.05 |
| + SSC | 45.8 | 45.5 | 43.9 | 46.1 | 20.9 | 20.7 | 21.2 | 20.6 | 21.4 | 20.6 | 21.8 | 33.09 |
| Titans (LMM) | 44.2 | 44.7 | 43.9 | 44.5 | 20.2 | 20.1 | 20.3 | 20.6 | 20.1 | 19.5 | 19.1 | 31.75 |
| + Log-Linear++ | 44.5 | 44.9 | 44.1 | 44.7 | 20.4 | 20.4 | 20.5 | 20.7 | 21.5 | 19.8 | 20.4 | 34.37 |
| + GRM | 50.2 | 47.5 | 45.3 | 50.9 | 21.7 | 21.8 | 21.9 | 21.5 | 23.7 | 23.4 | 23.3 | 40.50 |
| + Memory Soup | 48.3 | 46.6 | 44.8 | 49.4 | 21.3 | 21.4 | 21.7 | 21.1 | 22.9 | 22.2 | 22.5 | 38.43 |
| + SSC | 46.1 | 45.7 | 44.3 | 46.9 | 20.8 | 20.7 | 21.2 | 20.9 | 21.9 | 20.4 | 21.5 | 36.27 |

the models on short and long context. Additional details about the experimental setups are in Appendix C.

## 4.1. Language Modeling

We start with common academic-scale language modeling. The results of SWLA, DLA, and Titans with and without memory caching are reported in Table 1. There are three observations: (1) Comparing DLA, Titans, and SWLA with their enhanced version with memory caching, we observe that all memory caching variants provides consistent improvements on different downstream tasks, and also on average over their baseline. This shows the importance of memory caching to further enhance memory bounded models. (2) As discussed earlier, memory caching can be seen as a hybrid of (sparse) attention with recurrent models. Comparing memory caching enhanced models and attention-based models (i.e., hybrid and Transformers), memory caching provides a more powerful solution to the problem of limited memory in recurrent models. Particularly, Titans + MC and DLA + MC achieves +0.8% performance gain over the Titans. (3) Comparing MC's constant size segmentation and Log-Linear++ method, we observed that constant size segmentation variants provide better results. Furthermore, GRM and then SSC achieves better results among our provided methods. We attribute this performance gain to larger effective memory size that MC provides for the model.

## 4.2. Needle-In-A-Haystack Tasks

We evaluate the impact of MC on long-context retrieval using Needle-in-a-Haystack (NIAH) tasks (Table 2). MC-enhanced DLA and Titans consistently outperform the base models. Furthermore, MC variants outperform the Log-Linear approach, especially at longer contexts. Log-Linear struggles because it forces a single memory to compress very large initial segments (e.g., 8K tokens in a 16K sequence), whereas MC distributes the load more effectively.

## 4.3. In-context Retrieval Tasks

In-context recall tasks are among the most challenging benchmarks for recurrent neural networks. In this section, we follow Arora et al. (2024b) and perform experiments on SWDE (Lockard et al., 2019), NQ (Kwiatkowski et al., 2019), DROP (Dua et al., 2019), FDA (Arora et al., 2023), SQUAD (Rajpurkar et al., 2016), and TQA (Kembhavi et al., 2017) to evaluate and compare the performance of MC-enhanced variants with baselines and Transformers. The results are reported in Table 3. While Transformers still achieve the best results in in-context recall tasks, our MC variants show competitive performance, close the gap with Transformers, and performs better than state-of-the-art recurrent models. We again attribute this performance to larger memory capacity that scales with sequence length.

*Table 4.* Accuracy on LongBench tasks (Bai et al., 2024): NarrativeQA, QasperQA, MultiFieldQA, HotpotQA, 2WikiMultiQA, Musique, GovReport, QMSum, MultiNews, TREC, TriviaQA, SamSum, LCC, and RepoBench-P.

| Model | Single-Doc QA | | | Multi-Doc QA | | | Summarization | | | Few-shot | | | Code | |
|---|---|---|---|---|---|---|---|---|---|---|---|---|---|---|
| | NQA | QQA | MFQ | HQA | 2WM | Mus | GvR | QMS | MNs | TRC | TQA | SSM | LCC | RBP |
| Transformer | 11.5 | 9.6 | 19.1 | 21.5 | 28.9 | 6.5 | 13.0 | 9.2 | 3.1 | 27.2 | 27.9 | 15.1 | 22.9 | 29.1 |
| DLA | 9.4 | 17.5 | 12.1 | 11.8 | 22.3 | 4.8 | 9.5 | 7.4 | 5.1 | 4.8 | 23.5 | 9.7 | 38.4 | 34.9 |
| + Log-Linear++ | 10.1 | 10.2 | 17.1 | 12.4 | 23.3 | 5.5 | 6.6 | 12.7 | 5.8 | 18.6 | 24.7 | 16.2 | 31.6 | 31.0 |
| + GRM | 11.6 | 10.3 | 19.8 | 18.2 | 26.9 | 6.4 | 13.5 | 14.1 | 6.9 | 25.7 | 28.2 | 18.3 | 32.7 | 33.9 |
| + Memory Soup | 11.2 | 10.3 | 19.5 | 16.7 | 25.1 | 6.3 | 11.2 | 13.8 | 6.2 | 22.5 | 26.9 | 17.7 | 32.3 | 33.5 |
| + SSC | 10.7 | 10.2 | 18.8 | 14.2 | 24.8 | 5.9 | 8.4 | 12.9 | 6.1 | 20.5 | 25.7 | 16.8 | 31.9 | 32.6 |
| Titans (LMM) | 8.7 | 12.5 | 18.4 | 15.6 | 26.1 | 6.7 | 10.5 | 12.6 | 11.8 | 37.1 | 26.2 | 24.5 | 31.3 | 31.4 |
| + Log-Linear++ | 9.6 | 8.9 | 19.3 | 18.7 | 26.9 | 6.8 | 6.7 | 12.9 | 2.8 | 11.2 | 42.7 | 25.0 | 29.5 | 29.7 |
| + GRM | 11.8 | 9.4 | 19.9 | 21.4 | 29.1 | 7.2 | 8.4 | 13.3 | 3.1 | 14.8 | 49.7 | 25.5 | 31.0 | 32.8 |
| + Memory Soup | 10.7 | 9.2 | 19.6 | 20.2 | 28.2 | 7.1 | 7.8 | 13.1 | 3.0 | 13.7 | 47.1 | 25.3 | 30.8 | 31.4 |
| + SSC | 9.9 | 9.1 | 19.4 | 19.8 | 27.5 | 6.9 | 7.1 | 13.0 | 2.8 | 12.5 | 44.8 | 25.2 | 29.9 | 30.8 |

## 4.4. Long Context Understanding Tasks

We evaluate long-context understanding on LongBench (Bai et al., 2024) (Table 4). All MC-enhanced variants provide performance gains compared to their base RNNs, again

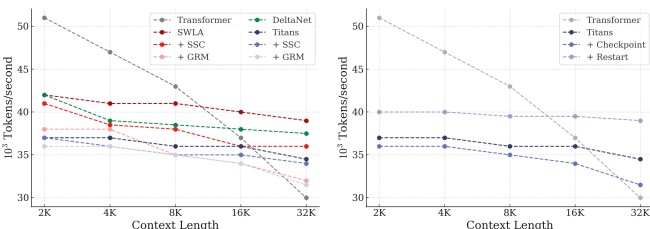

*Figure 3.* Training throughput comparison of memory caching variants and baselines.

attributed to their increased memory capacity.

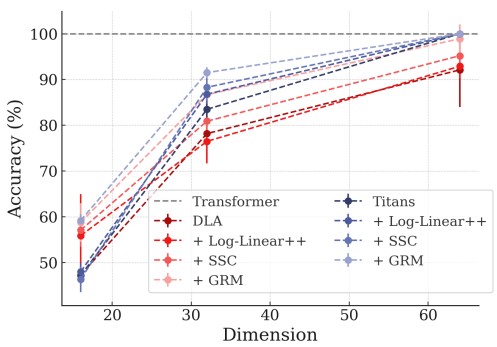

*Figure 4.* Average accuracy on MQAR over 5 seeds.

### 4.5. Multi-Query Associative Recall (MQAR)

In this section, we evaluate the performance of MC-enhanced variants in Multi-Query Associative Recall (MQAR) task (Arora et al., 2024a). The results are reported in Figure 4. Our models show good performance compared to their base RNNs also the state-of-the-art recurrent models, achieving the best performance per dimension value compared to state-of-the-art models such as Atlas (Behrouz et al., 2025a).

*Table 5.* Ablation Study on MC. All design choices of MC are positively contributing to its effectiveness.

| Model | Language Modeling ppl ↓ | C.S. Reasoning acc ↑ | Retrieval acc ↑ |
|---|---|---|---|
| Titans (GRM) | 13.3 | 58.3 | 40.5 |
| - Context-dependent | 13.4 | 57.4 | 33.0 |
| - Gating | 13.5 | 56.9 | 32.4 |
| - Linear Memory | 13.7 | 56.3 | 34.5 |
| - Shared $u$ and $q$ | 13.4 | 57.9 | 36.6 |
| Titans (SSC) | 13.4 | 57.6 | 36.3 |
| - Context-dependent | 13.4 | 57.1 | 32.6 |
| - Gating | 13.5 | 56.8 | 31.9 |
| - Linear Memory | 13.8 | 56.8 | 33.4 |
| - Shared $u$ and $q$ | 13.5 | 57.3 | 34.8 |
| Titans + Log-Linear | 13.5 | 56.4 | 32.8 |
| Titans + CDSA | 13.7 | 57.4 | 38.8 |

### 4.6. Ablation Studies

Next, we evaluate the effect of design choices in the MC framework. The first choice is wether $\gamma$ should be the function of only input or also the context of blocks. The results are reported in Table 5. This design choice has shown significant improvement on average. The second design is to remove the gating. Note that without gating, the design collapses into residual memory. The results show even this simple design can enhance the performance of the models. Finally, in the third design, we use a linear memory module. Surprisingly, using memory caching results in more robustness of the performance with respect to the memory architecture and expressivity.

**The Effect of Segment Size.** We further vary the segment size to evaluate its effect on the performance. In language modeling tasks, we vary segment size in $\{8, 32, 64, 128\}$, obtaining 16.38, 14.91, 15.26, and 17.09 as the perplexity of the models, respectively.

### 4.7. Efficiency

Finally, we evaluate the training throughput of our variants with baselines. The results are reported in Figure 3. Our MC variants provide a middle ground between Transformers and RNNs, and they become extremely efficient compared to Transformers, when increasing the context length. These results indicate that our SSC variant preserves the best of Transformers (i.e., growing memory) as well as RNNs (i.e., constant cost per token), while performs on par or better compared to other variants in the diverse downstream tasks that we discussed earlier, they also add minimal overhead compare to their original base RNN variant. Furthermore, they show significantly better efficiency in longer sequences.

Comparing the two variants of checkpointing or independent compressors, we observed that while the performance of the checkpointing is better (58.3% vs. 57.2% for Titans(GRM) in language modeling tasks), the efficiency of independent compressors variant is significantly better, mainly due to sequence parallelization that unlocks with this variant (each segment for compression is treated independently).

## 5. Conclusion

We present Memory Caching, a simple technique applicable to all recurrent neural networks, that caches a subset of memory state, allowing subsequent tokens directly attend to its past relevant tokens. Our experiments show improvements over a subset of baselines. A lot of choices in this paper have been made to keep the resulting model as simple as possible, better showing the effect of memory caching idea. However, in future work, more expressive pooling or routing mechanism can be used to further enhance the performance.

## Impact Statement

This paper presents work whose goal is to advance the field of machine learning. There are many potential societal consequences of our work, none of which we feel must be specifically highlighted here.

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

# A. Discussion and Proof of Concept

In this section we discuss the implication of using memory caching technique for linear and deep memory modules, and discuss the models we use as a proof of concept in our evaluations.

## A.1. Implication of Memory Caching on Linear and Deep Memory Modules

**Linear Memory.** Let us start with a simple but extreme case, where the segment size is 1 and also the recurrent memory is a vector-valued value-less memory module[1]. In this setup, each token $x_t \in \mathbb{R}^d$ is considered as a segment and so the memory state only stores this token: i.e.,

$$\mathcal{M}_1^{(t)} = \underbrace{\mathcal{M}_0^{(t)}}_{\mathbf{b_K}} + \boldsymbol{k}_t = \mathbf{b_K} + x_t \mathbf{W_K}, \tag{26}$$

where $\mathcal{M}_0^{(t)}$ acts as a bias term for the projection of input $x_t$ to key $\boldsymbol{k}_t$. Next, applying memory caching on this formulation results in:

$$\mathbf{y}_t = \sum_{i=1}^{t} \gamma_t^{(i)} \mathcal{M}_1^{(i)}(\mathbf{q}_t) = \sum_{i=1}^{t} \underbrace{\frac{\exp\left(\boldsymbol{u}_t^\top \boldsymbol{k}_i\right)}{\sum_{\ell=1}^{i} \exp\left(\boldsymbol{u}_t^\top \boldsymbol{k}_\ell\right)}}_{\gamma_t^{(i)}} \underbrace{\left(\mathbf{b_K} + x_i \mathbf{W_K}\right)}_{\mathcal{M}_1^{(i)}} \boldsymbol{q}_t \ . \tag{27}$$

Given the fact that $\boldsymbol{q}_t$ is a free parameter and a function of input, one can re-parametrize the above formulation by letting $\mathcal{M}_1^{(i)} = \boldsymbol{v}_i'$:

$$\mathbf{y}_t = \left(\sum_{i=1}^{t} \frac{\exp\left(\boldsymbol{u}_t^\top \boldsymbol{k}_i\right)}{\sum_{\ell=1}^{i} \exp\left(\boldsymbol{u}_t^\top \boldsymbol{k}_\ell\right)} \boldsymbol{v}_i'\right) \otimes \sigma\left(x_t \mathbf{W_Q}\right), \tag{28}$$

which is equivalent to gated global attention block–an enhanced variant of attention, where the output is gated with a linear projection of the input. Therefore, setting segment lengths equal to one and using a value-less vector-valued recurrent memory with memory caching, recover the gated global softmax attention block. This re-discovery, however, is based on the simplest design choices in the memory caching, which motivates designing architectures that are more powerful in principle. This design can be extended to more complex formulations of global attention with clear motivations. For example, the above formulation caches the direct value of token projection, meaning that each token is stored in a value-less memory module. One, however, can extend the above formulation by using a more expressive memory update: e.g., using linear attention (Katharopoulos et al., 2020).

The linearity of the memory and/or a simple retrieval process using multiplication (i.e., letting $\mathbf{y}_t = \mathcal{M}_t \boldsymbol{q}_t$), however, can cause unwanted simplification, resulting in collapsing the design or sub-optimal performance. For example, earlier in the paper, we discussed how memory fusion technique can collapse into (gated) residual memory variant for linear memories, while these two methods providing different solutions for deep memory modules. As an another example, in linear memory configuration, one can interpret retrieval process by $\boldsymbol{q}_t = x_t \mathbf{W_Q}$ as a gating architecture that is multiplied by the compressed context (i.e., linear memory). In this viewpoint, one can simplify the process by removing the gating and so set $\boldsymbol{q}_t = \mathbb{1} \in \mathbb{R}^d$. We refer to such modules as *compressors*.

Recently, hybrid models–i.e., architectures with interleaved global attention and memory-bounded modules (i.e., RNNs, sliding window attention, etc.)–have gained popularity, mainly due to their robust performance in recall intensive tasks while improving efficiency of pure global attention-based models. In particular, let us consider a compressor layer with a *vector-valued* recurrent memory, followed by a global attention block. Given $x_t$ as the input, the first block is computed as:

$$\boldsymbol{q}_t = \mathbb{1}, \qquad\qquad \boldsymbol{k}_t = x_t \mathbf{W_K}, \qquad\qquad \boldsymbol{v}_t = x_t \mathbf{W_V} \tag{29}$$

$$\mathcal{M}_t = f\left(\mathcal{M}_{t-1}; \boldsymbol{k}_t, \boldsymbol{v}_t\right), \qquad \mathbf{y}_t = \mathcal{M}_t \tag{30}$$

---

[1]Value-less memory module are associative memories whose mapping only depends on the keys. See Behrouz et al. (2025b) for the details.

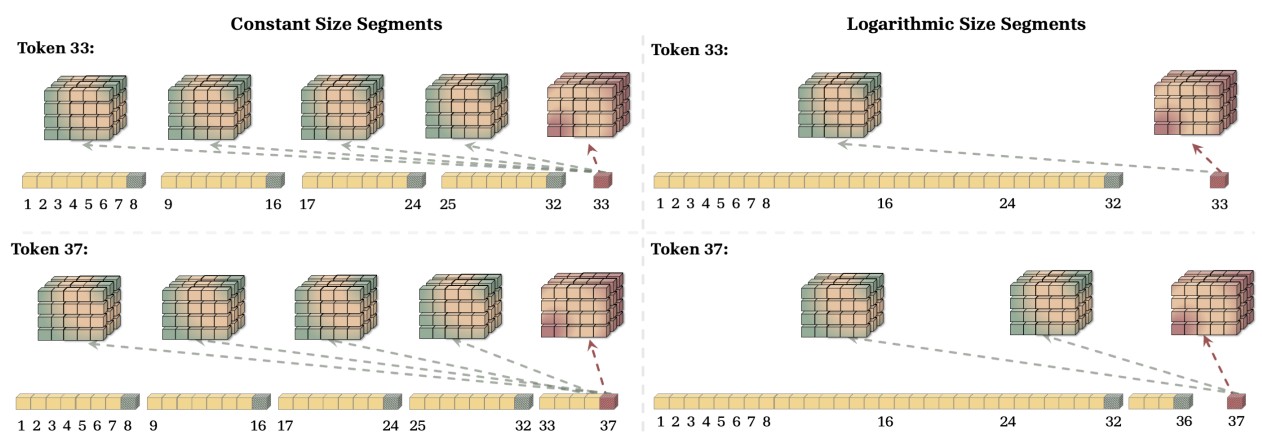

*Figure 5.* An illustrative example of memory caching with constant and logarithmic size segments. Logarithmic segmentation while computationally appealing, results in either long subsequences that might cause memory overflow, and/or short subsequences that prevents the memory to properly optimizes itself in the inner-loop.

Next, this output is considered as the input of the next block and the final output of the layer is computed as (we disregard normalizations for the sake of clarity):

$$q_t^{(\texttt{attn})} = \mathbf{y}_t \mathbf{W}_{\mathbf{Q}}^{(\texttt{attn})}, \qquad k_t^{(\texttt{attn})} = \mathbf{y}_t \mathbf{W}_{\mathbf{K}}^{(\texttt{attn})}, \qquad v_t^{(\texttt{attn})} = \mathbf{y}_t \mathbf{W}_{\mathbf{V}}^{(\texttt{attn})} \tag{31}$$

$$\mathbf{o}_t = \sum_{i=1}^{t} \frac{\exp\left(q_t^{(\texttt{attn})} k_i^{(\texttt{attn})}\right)}{\sum_{\ell=1}^{i} \exp\left(q_t^{(\texttt{attn})\top} k_\ell^{(\texttt{attn})}\right)} v_i^{(\texttt{attn})} \tag{32}$$

Note that $\mathbf{y}_t = \mathcal{M}_t$, and so, interestingly, the above formulation is equivalent to the memory caching with segment size of 1, when we cache memory *checkpoints* instead of using independent memories (see Section 3.4). This viewpoint not only provides a reason for why a famous recipe of hybrid models should work (because it can enhances the effective memory capacity of the recurrent model), but it also motivates designing more powerful variants by seeing attention as a module that enforces caching past inputs. Note that the above explanation demonstrates an equivalency only for an oversimplified version, and with considering normalization and feed-forward layers the modeling expressivity of both cases can be different. The above discussion, however, supports the power of memory caching technique and motivates going beyond $q_t = \mathbb{1}$. The case where $q_t = x_t \mathbf{W}_{\mathbf{Q}}$, memory caching can result in an ad-hoc attention, where the input sequence to attention block is different for each query $q_t$. That is, instead of fully pre-determined sequence of tokens as the input of attention block in hybrid variants (i.e., outputs of the last memory layer, $\mathbf{y}_t$), memory caching allows the model to build its own input sequence based on the query $q_t$ (i.e., $\{\mathcal{M}_{L^{(i)}}^{(i)}(q_t)\}_{i=1}^{s}$).

**Deep Memory.** Contrary to the above, memory caching with deep memory results in a new class of architectures with no overlap with their hybrid variants. Let us revisit the initial example we used for linear memory modules. We let memory $\mathcal{M}(\cdot)$ be a 2-layer MLP, and each token $x_t \in \mathbb{R}^d$ be a segment (i.e., segment size is 1 and so each memory only stores one token): i.e.,

$$\mathcal{M}_1^{(t)} = \mathcal{M}_0^{(t)} - \eta_t \nabla \mathcal{L}(\mathcal{M}_0^{(t)}; k_t, v_t). \tag{33}$$

This has two implications: (i) Now, each token is represented by a tensor rather than a matrix or vector. Therefore, a token is no longer represented by a constant vector and given the query $q_t$ the context and representation of token $x_i$ can be different (i.e., $\mathcal{M}_1^{(i)}(q_t)$); (ii) The initial point or bias term $\mathcal{M}_0^{(i)}$, which encodes general knowledge, is a neural network. Therefore, its effect on retrieval process of different queries (i.e., $\mathcal{M}_1^{(i)}(q_t)$) is not the same.

### A.2. The Effect of Segmentation

One of the critical design choices in memory caching is the segmentation of the sequence. Intuitively, segment lengths provides a trade-off between the level of compression and computational cost: E.g., (1) As discussed earlier, Transformers can be seen as memory caching with segment size of 1, meaning that each token itself is cached (no compression, high

computational cost of $\mathcal{O}(L^2)$); (2) An RNN module is the extreme case of memory caching, where the entire sequence is considered as a segment and only a single online memory is cached (full compression, constant computational cost per token, $\mathcal{O}(L)$).

In more details, for a token $x_t$, let us split the sequence into segments $S^{(1)}, \ldots, S^{(N)}$ with size $L^{(1)}, \ldots, L^{(N)}$ and use memories $\mathcal{M}^{(1)}, \ldots, \mathcal{M}^{(N)}$ to compress the segments. The compression process for each segment is linear with respect to the segment size, i.e., $\mathcal{O}(L^{(i)})$. Therefore, the memory update operation has a cost of $\mathcal{O}(\sum_{i=1}^N L^{(i)}) = \mathcal{O}(L)$ and the retrieval process requires forward pass over all past cached memories (one memory per segment). Therefore, assuming forward pass of memory requires $\mathcal{O}(p)$ operations, the computation cost *per token* would be $\mathcal{O}(p \times N)$. Therefore, the total cost for memory caching is $\mathcal{O}(L + p \times N \times L)$. Note that $N$, here, is the function of segmentation. As a example, if segments have equal lengths with size $C = \frac{L}{N} \geq 2$, then the computational cost is $\mathcal{O}(p \times \frac{L^2}{C})$, similar to Transformers but with a smaller constant term (i.e., better efficiency). Another example is to split the sequence logarithmically: i.e., writing $L$ in the binary format, and then let $L^{(i)}$ be the corresponding power of 2 to the indices of non-zero elements. Figure 5 (Left) shows an example of this segmentation. For a sequence length of $37 = (100101)_2$, this segmentation uses three segments with lengths $L^{(1)} = 32$, $L^{(2)} = 4$, and $L^{(3)} = 1$. Therefore, in this formulation, the maximum value of $N$ is $\log_2(L)$ and so the computational complexity is $\mathcal{O}(p \times L \log(L))$.

As discussed earlier, the segmentation provides a trade-off between the level of compression (recall performance) and computational cost. Figure 5 illustrates this trade-off by an example of constant-size and logarithmic segmentation. As discussed above, the time complexity of these methods are $\mathcal{O}(\frac{L^2}{C})$ and $\mathcal{O}(L \log(L))$, respectively, which makes the logarithmic segmentation more efficient. On the other hand, however, logarithmic segmentation provides a significantly less resolution for long past tokens, limiting its ability in recall-intensive tasks.

### A.3. Memory Caching for Titans and Linear Attention Variants

As discussed earlier in the paper, the memory caching technique is applicable to any arbitrary recurrent update rule and potentially can enhance their effective context length. As a proof of concept, we use use memory caching on the update rule of Sliding Window Linear Attention (SWLA) and Deep Linear Attention (DLA) (Behrouz et al., 2025a), Linear Attention (Katharopoulos et al., 2020), and Titans (Behrouz et al., 2024).

**Sliding Window Linear Attention.** Recently, Behrouz et al. (2025a) introduced Sliding Window Linear Attention (SWLA), in which the memory updates its weights based on a set of $c \geq 1$ past tokens (contrary to the online RNNs, where the memory is updated solely based on the last token). More specifically, given a memory module $\mathcal{M}(\cdot)$ and keys, values, and queries, $\{(\boldsymbol{k}_t, \boldsymbol{v}_t, \boldsymbol{q}_t)\}_{t=1}^L$, the update and retrieval rules are defined as:

$$\mathcal{M}_t = \alpha_t \mathcal{M}_{t-1} + \sum_{i=t-c+1}^t \beta_i^{(t)} \boldsymbol{v}_i \boldsymbol{k}_i^\top, \tag{34}$$

$$\boldsymbol{y}_t = \mathcal{M}_t \boldsymbol{q}_t, \tag{35}$$

When $c = 1$, the design collapses to the simple linear attention (online linear RNN) and its gated variants (Katharopoulos et al., 2020; Yang et al., 2024b; Li et al., 2025). As a proof of concept, we use memory caching on SWLA with $c = 2$, resulting in a recurrence and retrieval of:

$$\mathcal{M}_t^{(s)} = \alpha_t \mathcal{M}_{t-1}^{(s)} + \left( \beta_t \boldsymbol{v}_{t-1} \boldsymbol{k}_{t-1}^\top + \lambda_t \boldsymbol{v}_t \boldsymbol{k}_t^\top \right), \tag{36}$$

$$\boldsymbol{y}_t = \gamma_t^{(s)} \mathcal{M}_t^{(s)} \boldsymbol{q}_t + \sum_{i=1}^{s-1} \gamma_t^{(i)} \mathcal{M}_{L^{(i)}}^{(i)} \boldsymbol{q}_t. \tag{37}$$

Note that, as discussed earlier, since SWLA is a linear memory module, both GRM and Memory Soup variants result in the same formulation.

**Deep Linear Attention.** DLA uses the same update rule as linear attention (i.e., Hebbian rule), but with a deep memory module. That is, given a memory module $\mathcal{M}(\cdot)$ and keys, values, and queries, $\{(\boldsymbol{k}_t, \boldsymbol{v}_t, \boldsymbol{q}_t)\}_{t=1}^L$, the update and retrieval rules are defined as:

$$\mathcal{M}_t = \mathcal{M}_{t-1} - \eta_t \nabla \mathcal{L}(\mathcal{M}_{t-1}; \boldsymbol{k}_t, \boldsymbol{v}_t), \tag{38}$$

$$\boldsymbol{y}_t = \mathcal{M}_t(\boldsymbol{q}_t), \tag{39}$$

where the attentional bias objective is defined as $\mathcal{L}\left(\mathcal{M}_{t-1}; \boldsymbol{k}_t, \boldsymbol{v}_t\right) = -\langle\mathcal{M}_{t-1}(\boldsymbol{k}_t), \boldsymbol{v}_t\rangle$. Using memory caching (GRM variant), the update and retrieval process for DLA are defined as:

$$\mathcal{M}_t^{(s)} = \mathcal{M}_{t-1}^{(s)} - \eta_t \nabla\mathcal{L}\left(\mathcal{M}_{t-1}^{(s)}; \boldsymbol{k}_t, \boldsymbol{v}_t\right), \text{ for } 1 \leq t \leq L^{(s)}, \tag{40}$$

$$\mathbf{y}_t = \gamma_t^{(s)} \mathcal{M}_t^{(s)}(\mathbf{q}_t) + \sum_{i=1}^{s-1} \gamma_t^{(i)} \mathcal{M}_{L^{(i)}}^{(i)}(\mathbf{q}_t). \tag{41}$$

Similarly, we can replace Equation 12 or Equation 14 with the update rule of DLA (similar to Equation 40) to derive other memory caching variants. Note that when memory module $\mathcal{M}(\cdot)$ is a matrix, then the above formulation is equivalent to the linear attention (Katharopoulos et al., 2020).

**Titans.** In Titans, compared to DLA, both attentional bias objective as well as the internal optimizer are different. More specifically, given a memory module $\mathcal{M}(\cdot)$ and keys, values, and queries, $\{(\boldsymbol{k}_t, \boldsymbol{v}_t, \boldsymbol{q}_t)\}_{t=1}^L$, the update and retrieval rules of Titans are defined as:

$$\mathcal{M}_t = \alpha_t \, \mathcal{M}_{t-1} - \mathcal{S}_t, \tag{42}$$
$$\mathcal{S}_t = \beta_t \, \mathcal{S}_{t-1} - \eta_t \nabla\mathcal{L}\left(\mathcal{M}_{t-1}; \boldsymbol{k}_t, \boldsymbol{v}_t\right), \tag{43}$$
$$\mathbf{y}_t = \mathcal{M}_t(\boldsymbol{q}_t), \tag{44}$$

where the attentional bias objective is defined as $\mathcal{L}\left(\mathcal{M}_{t-1}; \boldsymbol{k}_t, \boldsymbol{v}_t\right) = \|\mathcal{M}_{t-1}(\boldsymbol{k}_t) - \boldsymbol{v}_t\|_2^2$. With MC, while the memory update operation for each segment is the same as Equation 42 and Equation 43, the retrieval process for Titans with memory caching is defined the same as Equation 41.

**Log-Linear++ Variant.** Recently, Guo et al. (2025) take advantage of structured matrix formulation of linear RNNs and design log-linear attention, a hierarchical algorithm based on Fenwick tree structure (Fenwick, 1994) that allows a logarithmically growing set of hidden states. We aim to use log-linear attention as the baseline in our experiments to show the effect of segmentation on the efficiency and retrieval performance. Its formulation, however, suffer from the positional bias and lack of context-dependency in retrieval process that we discussed in Section 3.1. For the sake of fair comparison, we improve Log-linear formulation, denoted as Log-Linear++ in our experiments, by reformulating it as a variant of memory caching with GRM and logartithmic-size set of segments. The segmentation process is the same as the process describe in Section A.2. The other components are kept unchanged, the same as our memory caching variants.

# B. Related Work

## B.1. Modern Recurrent Neural Networks

Recent efforts have focused on alleviating the quadratic complexity and context-length limitations of Transformers, motivating the development of efficient recurrent alternatives that offer faster inference and training (Tiezzi et al., 2024). Early models such as RetNet (Sun et al., 2023), RWKV (Peng et al., 2023), and S5 (Smith et al., 2023) relied on data-independent transition matrices with Hebbian-like updates. A subsequent wave of work introduced input-dependent parameters into linear RNN architectures (e.g., linear RNNs (Hasani et al., 2023; Smith et al., 2023; McDermott et al., 2025), RWKV6 (Peng et al., 2024)), alongside more expressive update mechanisms, including variants of the delta rule (Peng et al., 2025; Schlag et al., 2021; Yang et al., 2024a; Liu et al., 2024). This line of research has been further extended to deeper architectures, incorporating delta-rule-like updates (Irie et al., 2021; Sun et al., 2024) or data-dependent momentum-based rules with forget gating (Behrouz et al., 2024). More recently, Siems et al. (2025) enhanced delta-rule models by applying multiple gradient descent updates per token, yielding more expressive state-tracking capabilities. Beyond linear recurrent models, several works investigate RNNs with non-linear recurrence (Csordás et al., 2024; Merrill et al., 2024; Lim et al., 2024; Behrouz et al., 2025b;a; Schöne et al., 2025; Karami & Mirrokni, 2025; Von Oswald et al., 2023; Gonzalez et al., 2024; Behrouz et al., 2025c), with emphasis on accelerating their training (Gonzalez et al., 2024; Lim et al., 2024; Schöne et al., 2025; Li et al., 2026).

**Fast Weight Programs and Meta Learning.** The view of linear layers as key-value associative memory systems dates back to Hopfield networks (Hopfield, 1982). This idea was later extended through the development of fast weight programmers, in which dynamic fast programs are integrated into recurrent neural networks to function as writable memory stores (Schlag

*Table 6.* Architectural Details.

| Model | Block | Dim | Head | Peak LR | Token |
|-------|-------|------|------|---------|-------|
| 760M  | 24    | 1536 | 16   | 1.25e-3 | 30B   |
| 1.3B  | 18    | 2048 | 8    | 7e-4    | 100B  |

et al., 2021; Schmidhuber, 1992; 1993). Among the learning paradigms for such systems, Hebbian learning (Hebb, 2005) and the delta rule (Prados & Kak, 1989) have been most prominent. Both rules have been extensively studied in the literature (Munkhdalai & Yu, 2017; Schmidhuber, 1992; Munkhdalai et al., 2019; Schlag et al., 2021; Irie et al., 2021; Yang et al., 2024c;a).

**Hopfield Networks.** Our formulation builds on the broad concept of associative memory, where the goal is to learn mappings between keys and values. Seminal work by Hopfield (1982) introduced Hopfield Networks as one of the earliest neural architectures explicitly based on associative memory, formalized through the minimization of an energy function for storing key-value pairs. While classical Hopfield networks have seen reduced applicability due to limitations in vector-valued memory capacity and the structure of their energy function, recent studies have sought to enhance their capacity through various approaches (Krotov, 2021; Li et al., 2024; Krotov & Hopfield, 2016). In particular, extensions of their energy functions with exponential kernels have been explored (Krotov & Hopfield, 2016; Lucibello & Mézard, 2024). Moreover, connections between modern Hopfield networks and Transformer architectures have been actively investigated (Ramsauer et al., 2021; Hu et al., 2024).

### B.2. Efficient Attention Mechanisms

In addition to recurrent architectures, recent work has proposed using structured matrices to improve the efficiency of token and channel mixing layers. For example, Butterfly matrices (Dao et al., 2019), Monarch matrices (Dao et al., 2022), and Block Tensor-Train matrices (Qiu et al., 2024) provide compact yet expressive parameterizations that reduce the computational burden of dense projections. Other approaches design sparse or hybrid attention mechanisms, such as sliding-window attention or models that combine localized recurrence with selective long-range connections (Nguyen et al., 2021; Arora et al., 2024b; Munkhdalai et al., 2024). Another family of approaches reduces the quadratic complexity of attention to nearly log-linear time. Classical examples include Reformer (Kitaev et al., 2020), which uses locality-sensitive hashing to cluster queries and keys, and LogSparse Transformer (Li et al., 2019) and Informer (Zhou et al., 2021), which rely on structured sparsity patterns for efficiency in long-sequence and time-series tasks. Subsequent work has introduced more elaborate designs, such as multi-resolution attention (Zeng et al., 2022), which progressively refines attention scores from coarse to fine levels, and Fast Multipole Attention (Kang et al., 2023), which adapts the fast multipole method for scalable long-range interactions. Recently, Guo et al. (2025) introduce Log-Linear Attention, a framework that augments linear attention with a logarithmically growing set of hidden states organized via Fenwick tree partitioning. This design achieves $\mathcal{O}(L \log L)$ training complexity and $\mathcal{O}(\log L)$ decoding memory, while preserving hardware-efficient parallelization.

## C. Experimental Details

In our experimental setup we follow recent studies on recurrent models (Yang et al., 2024a; Behrouz et al., 2024; 2025b; Zhang et al., 2025; Guo et al., 2025), we use Wikitext (Merity et al., 2017), LMB (Paperno et al., 2016), PIQA (Bisk et al., 2020), HellaSwag (Zellers et al., 2019), WinoGrande (Sakaguchi et al., 2021), ARC-easy (ARC-e) and ARC-challenge (ARC-c) (Clark et al., 2018), SIQA (Sap et al., 2019), and BoolQ (Clark et al., 2019). In the training, we use a vocabulary size of 32K and use training length of 4K-32K tokens. We employ AdamW optimizer with learning rate of $4e$-$4$ with cosine annealing schedule with batch size of 0.5M tokens, and weight decay of $0.1$. For the memory architecture, unless state otherwise, we use an MLP with 2 layers with expansion factor of 4 and GELU activation function (Hendrycks & Gimpel, 2016). We also use residual connections and layer norm at the end of each chunk: $\mathcal{M}(x) = x + W_1 \sigma(W_2 x)$.

