# OpenReview forum: "Memory Caching: RNNs with Growing Memory"
_ICML.cc/2026/Conference — ICML 2026 regular_

### Official Review · Reviewer_HUyF · 2026-03-02

**Soundness:** 2
**Presentation:** 3
**Significance:** 3
**Originality:** 3
**Overall Recommendation:** 3
**Confidence:** 3

**Summary:**

This paper proposes a memory caching technique to address the recall bottleneck caused by the fixed memory capacity of recurrent neural networks by caching the hidden states at the end of sequence segments. This method allows the RNN's memory to grow with the sequence length, thus providing an adjustable balance between the O(L) complexity of standard RNNs and the O(L^2) complexity of Transformers, with a typical complexity of O(N*L). The authors introduce several aggregation mechanisms: residual memory, gated residuals, a memory soup of parameter space, and sparse selective caching based on Top-k routing. Experiments on architectures such as Titans, DLA, and SWLA validate the effectiveness of this technique in language modeling, long text understanding, and needle in a haystack tasks.

**Compliance With Llm Reviewing Policy:**

Affirmed.

**Final Justification:**

My concerns are partially resolved, I will keep my score.

**Key Questions For Authors:**

1. Regarding the actual hardware efficiency of the SSC variant, although it theoretically reduces complexity to O(k*L), considering the overhead of routers and the underlying cost of frequently loading high-dimensional memory checkpoints (hidden states) for each token, the bottleneck of this design on real hardware may be very significant. Could the authors provide a detailed latency analysis report comparing the proportion of memory retrieval and loading time to actual computation time?

2. Since the core components of MC, such as MoE-style routing for SSC and weighted soup techniques for Memory Soup, have mature applications in other fields, besides empirically applying these techniques to RNN hidden states, could the authors clarify whether there is a unique theoretical or architectural insight specifically for the evolutionary characteristics of recurrent models? In particular, why is the interpolation parameter (Memory Soup) more efficient than simple output aggregation in the context of "optimizing checkpoints"?

3. The choice of fragment length C is clearly a key hyperparameter determining the compression efficiency tradeoff, but the ablation experiments in Section 4.6 seem too weak. Could this parameter be supplemented with some ablation experiments? Could the authors demonstrate the dynamic relationship between model performance and inference latency as C varies from 1 to L?

4. In Memory Soup, you interpolated the weights of the deep memory modules. Given that the weight space of neural networks is typically non-convex, why doesn't averaging these "optimization checkpoints" lead to representation collapse or deviation from the learned manifold? Are there any stability analyses?

5. The paper positions MC as an intermediate zone between standard recurrent and attention architectures, but what are the advantages of this design compared to standard hybrid architectures, such as inserting a global attention layer every N layers of an RNN?

**Limitations:**

Yes. The trade-offs between memory compression and computational cost and the performance delta between checkpointing and independent compressors

**Strengths And Weaknesses:**

Strengths:

1. The authors provide extensive experimental validation. This research validates its generality on various modern sub-quadratic architectures, including Titans, Deep Linear Attention, and Sliding Window Linear Attention. Experimental benchmarks cover language modeling, commonsense reasoning, LongBench, and challenging "needle in a haystack" tasks, demonstrating the model's performance improvements across different dimensions.

2. The paper cleverly utilizes the perspective of "test-time optimization," treating hidden states as dynamic entities in the optimization process. This perspective provides reasonable logical support for the design of memory caching, i.e., optimization checkpoints, making the technical solution intuitively easy to understand.

3. The proposed Memory Caching (MC) framework achieves interpolation between the O(L) complexity of standard RNNs and the O(L^2) complexity of Transformers by adjusting fragment lengths, providing practical engineering flexibility.

Weaknesses:

1. Insufficient originality and architectural innovation. The technical roadmap features highly reusable components, with core variants largely being direct transfers of existing mature technologies. For example, the Memory Soup variant directly borrows the weighted averaging idea from Weight Soup; while Sparse Selective Caching borrows the Top-k routing logic from Mixture of Experts. This results in limited depth of combinatorial innovation: the work is more engineering-oriented and lacks original mathematical insights into the evolutionary characteristics of hidden states in recurrent models.

2. Insufficiently detailed ablation experiments. The paper's analysis of the impact of the core hyperparameter fragment length is too simplistic, only showing a small change in perplexity. There is a lack of Pareto front analysis regarding the relationship between fragment size and actual inference latency and memory usage. The ablation experiments in Table 5 fail to adequately isolate the specific contributions of various complex components (such as different pooling functions or routing mechanisms) to the final recall performance.

3. The paper needs more interpretability and in-depth analysis. While the paper considers caching as "optimization checkpoints," it fails to provide empirical analysis demonstrating which information these snapshots capture that the underlying RNN cannot retain. In the SSC variant, the logic behind the router's selection of historical segments lacks qualitative visualization, failing to explain why the model selects specific segments for review within long sequences.

4. The paper only shows throughput in Figure 3 but does not discuss the memory bandwidth pressure and I/O latency caused by SSC's frequent retrieval and loading of large amounts of historical states on real-world hardware (such as GPUs).

---

> ### Author Rebuttal · Authors · 2026-03-31
>
> Thank you for your detailed and constructive review. Following your suggestions, we have added additional experiments to address your questions, and as a result we believe these changes have strengthened our paper. Please find our detailed responses below:
>
> > Insufficient originality and architectural innovation
>
> **Response:**
> We respectfully disagree with the reviewer that this work lacks originality. A part of science has always been about the use of known tools/knowledge in new contexts/concepts/tasks. The idea of caching memory states in the recurrent neural networks, which results in a growing effective memory, and also designing residual connection across the sequence dimension is a novel direction and idea. This idea, in fact, is addressing a long standing challenge in recurrent models: i.e., having a fixed-size memory.
>
> The concept of mixture of expert, or weight averaging are very general techniques that are explored in other concepts/spaces/areas and to the best of our knowledge have never been used in similar cases. Considering a work as not original because it uses such general idea/techniques can exclude a lot of studies and research papers such as mixture of expert across depth, width, attention computation, etc. The novelty is to understand such tools can be useful in a new dimension and context.
>
> Please note that the gating we use for combining memory states is another purely novel idea, to the best of our knowledge. In fact, most gating mechanism are either input-independent or dependent, which is often defined as the linear projection of the current token. Our gating, however, (as motivated in Sec. 3.1) not only depends on the current token, but it also incorporates a context-dependency of the past. In our ablation in Table 5, we show that this design is very important particularly for retrieval and long-context tasks.
>
>
> >  Detailed ablation experiments on Segment Size
>
> **Response:** Please see the results in our response to `Reviewer Aiit`
>
> Also, please kindly note that this is an academic research paper with limited budget, and your requested ablation (i.e., ranging C from 1 to L) requires training 4096 independent models, which at least need about 7M dollors cost for training. We believe the current ablation is conclusive as the pattern is: with increasing the segment size, the performance gets better until some point and then increasing the segment size start to damage the performance. We have provided additional results on segment size and also checkpointing vs. independent memory.
>
> > Interpretability study and analysis
>
> **Response:** Thank you for your suggestion. Unfortunately, there is no revision opportunity, but following your suggestion, we have performed a new experiment, visualizing the value of gates over the past cached memory states, showing how internally MC retrieve the data from the past based on the tokens. Mainly the results support the fact that based on the query, the model set non-zero weights to the past cached memory states, resulting in performance improvement.
>
>
> > Advantages of this design compared to standard hybrid architectures
>
> **Response:** Please note that we have a dedicated section with several discussions on the connection to hybrid models in Appendix A. Note that, our method is still considered an attention free approach (i.e., pure RNNs) and can be seen as a path that could address the fixed-size memory issue of RNNs and so go towards purely attention free architectures in a longer term.
>
>
> > Why doesn't averaging checkpoints lead to representation collapse
>
> **Response:** We understand the importance of this question and acknowledge that we are not providing an answer to this question. However, please note that: (1) Our paper has a clear message that the memory of recurrent models can grow similar to attention and this can be done by caching the memory states over the sequence. We support this by our empirical evaluations showing that MC-enhanced models consistently perform better than their base RNNs. (2) Even well-known methods and concepts after hounders or even thousands of follow up studies are still being evaluated, understood, and studied. We hope different aspects of our MC, including the questions about why non-convexity does not cause any issue can be studied more in the future follow up works.
>
> As a general note, please note that the same issue raised by the reviewer can be said for weight souping methods. Indeed it is an interesting question, and deserves its own study, time, and exploration, beyond this paper.
>
>
> > Actual hardware efficiency of the SSC variant
>
> **Response:** Please note that the hardware efficiency of such methods can heavily rely on the implementation and specialized kernels, which is out of the scope of our work. The training throughput and also the scale of our experiments support the fact that these methods can reasonably run on hardware accelerators and show competitive throughput compared to other baselines.

---

> > ### Author Rebuttal · Reviewer_HUyF · 2026-04-02
> >
> > I would like to thank the authors for their hard work and the comprehensive details provided in the rebuttal. The author's rebuttal successfully addressed several key concerns by providing an Interpretability Study through gate value visualizations, clarifying the Relationship to Hybrid Architectures in Appendix A, reinforcing the Novelty of the Gating Mechanism, and supplying Additional Empirical Data on segment sizes and memory checkpointing.
> >
> > However, despite these strengths, several concerns remain unresolved or were partially misinterpreted during the rebuttal:
> >
> > Ablation Study on Segment Size C: There is a significant disconnect regarding the request for a more granular analysis of the segment size C. The authors framed the need for a deeper ablation as an impractical demand for an exhaustive sweep of 4,096 models costing $7 million.
> >
> > And it's better if has stability analysis and SSC efficiency analysis.

---

> > > ### Author Response · Authors · 2026-04-08
> > >
> > > Thank you very much for your time and constructive comments. We really appreciate it.
> > >
> > >
> > > We apologize for the misunderstanding. Following your suggestion, we have added a table with more values of segment size and provided both effectiveness and efficiency of the model based on different values of C to better understand the dynamic:
> > >
> > > | Segment Size | Performance | Efficiency |
> > > | - | - | - |
> > > C = 4|	16.27|	41|
> > > C = 8|	16.38|	39|
> > > C = 32|	14.91|	38|
> > > C = 64|	15.26|	38|
> > > C = 128|	17.09|	37|
> > > C = 256|	17.4|	38|
> > > C = 512 |   17.51   |    37 |
> > > C = 4096|   17.78 | 41|
> > >
> > > Also, as you suggested, we have provided the efficiency analysis of SSC in the following table:
> > >
> > > | \#blocks | Efficiency |
> > > | - |  - |
> > > | 1 | 41 |
> > > | 4 | 40 |
> > > | 8 | 40 |
> > > | 16 | 39 |
> > > | 64 |  38 |
> > >
> > > Please note that SSC show promising performance and efficiency, but still there is room for improvement in future by designing efficient hardware-aligned algorithms for (off)loading the experts/blocks.
> > >
> > > We hope our responses have answered your questions. We would really appreciate it if you could kindly consider re-evaluating the score of our paper. Thank you again for your time and for engaging with us during the rebuttal phase.

---

### Official Review · Reviewer_kc99 · 2026-03-09

**Soundness:** 3
**Presentation:** 3
**Significance:** 3
**Originality:** 2
**Overall Recommendation:** 4
**Confidence:** 4

**Summary:**

This paper introduce Memory Caching(MC), to improve the recurrent or linear models with a linear growing external memory. The core idea is to partition the input into segments and cache the memory by segments' last memory state, and allow the current token to read not only from the online memory but also the previous relative memory state from the whole cached memory.
The paper presents several variants of its main idea: residual aggression, gated residual memory (GRM), Memory Soup, and sparse selective caching(SSC). By evaluate on several recurrent and linear backbone on different benchmarks, it shows that the memory caching can improve the base model on long-context retrieval and recall tasks.

**Compliance With Llm Reviewing Policy:**

Affirmed.

**Final Justification:**

I believe the authors can make the appropriate revision based on the rebuttal.

**Key Questions For Authors:**

1 Can authors provide either direct comparisons to efficient long-context alternatives such as InfLLM-v2 and DSA?

2 How do the authors position MC relative to recent methods such as RESONA and Overcoming Long-Context Limitations of State-Space Models via Context-Dependent Sparse Attention?

3 Can you compare to Atlas on MQAR and your other benchmarks.

**Limitations:**

See weakness and questions.

**Strengths And Weaknesses:**

Strengths:
1. The motivation is good, and the problem is important: how to improve the long-context memory capacity of recurrent/linear-memory models.
2. The memory update method is intuitive and can be a useful for both training and inference.
3. The retrieve/selection method is easy but fast, and good for parallel since it do not influence the parallel design.
In that sense, the paper has clear potential significance.

Weakness:

1. The paper does not compare with strong enough similar complexity baseline. It is good to see consistent gains over base model, but in practice the more relevant questions is when one should use MC instead of other long-context approaches. It would be better to have compare with InfLLM-V2 and DSA such sparse-attention style methods. Without that, it is hard to judge the practical advantages of MC.
2. I found that the author claim it compare with Atlas on MQAR, but I did not see the results.
3. More related recent work should be discussed. In particular, RESONA and Overcoming Long-Context Limitations of State-Space Models via Context-Dependent Sparse Attention seem very relevant, since they also try to address the long-context limitations of linear / recurrent model, and use selected compressed segments.
4. The mean pooling methods may not good at complex multi-hop QA if the relative chunks can not be select by the semantic pooling.

---

> ### Author Rebuttal · Authors · 2026-03-31
>
> Thank you for your detailed and constructive review. Following your suggestions, we have added additional experiments to address your questions, and as a result we believe these changes have strengthened our paper. Please find our detailed responses below:
>
> > Compare with strong enough similar complexity baseline such as DSA and InfLLM-V2.
>
> **Response:** Please note that studies such as DSA and InfLLM-V2 are completely orthogonal to our work, and somehow irrelevant. In fact, these works have studied methods that can make **softmax global attention** more efficient and so unlocking the processing of longer sequences by **Transformers**. On the other hand, the goal of our study is to overcome the fixed-size memory state of **recurrent neural networks (RNNs)** and how to close the performance gap between them and Transformers. Therefore, the comparison of these methods is not an apple-to-apple comparison. If there is a specific message that the reviewer believes this comparison can bring, we would be happy to clarify or compare.
>
> Please note that, the goal of memory caching is not to achieve state-of-the-art performance for long-context tasks. Indeed, there are many methods that can be combined to further improve the performance. The main goal here, however, is to show RNNs can have growing memory similar to Transformers and so it results in closing the gap between the performance of RNNs and Transformers.
>
> As the standard practice in the literature, we also kindly refer to the very recently accepted paper Log Linear Attention (ICLR 2026), which is the closest work to ours and also its extended and more powerful version is used as the baseline in our experiments.
>
>
>
> > Compare with Atlas on MQAR and other benchmarks
>
> **Response:** Thank you and we are sorry for this misunderstanding. The DLA and SWLA baselines are from Atlas paper and are variants presented in that work. What we meant for MQAR was having DLA as the baseline. Please note that MC is a method that needs to be added to an update rule, and in all experiments, we compare an update rule with or without MC to show how it can improve the performance of fixed-size memeory models. We have already evaluated the effect of MC on Atlas (i.e., Omega update rule) in SWLA and also DLA, which are a linear attention with Omega and Hebbia-rules, respectively (both presented in Atlas paper).
>
>
> > Discussion on RESONA and CDSA:
>
> **Response:** The goal of RESONA and CDSA are different from MC and even are orthogonal, meaning that they can be applied together. Please note that our goal is to show that the memory of RNNs can grow with sequence length and this is beneficial, in the sense that the performance of the base model can be improved by letting the memory grow. The goal of the mentioned studies are to enhance the performance of fixed-size memory recurrent models by: (1) managing the model's input to integrate retrieved information from the provided context; and (2) using hybrid models that combines RNNs with context-dependent sparse attention. Please note that these techniques are general and are independent of the architecture. MC, on the other hand, directly target the architecture, and so an MC-enhanced model can further be augumentd using these techniques.
>
> However, following your suggestion, here there is a result of MC- and CDSA-enhance Titans:
>
> | Model | Language Modeling (lower better) | C.S. Reasoning (higher better)  | Retrieval (higher better) |
> |-|-|-| - |
> Titans + GRM |  13.3 | 58.3 | 40.5 |
> Titans + CDSA | 13.7 | 57.4 | 38.8 |
>
> MC-enhanced variant perform better than CDSA.
>
>
> > The mean pooling methods may not good at complex multi-hop QA
>
> **Response:** Please note that even in the rare failur scenarios, MC-enhanced models can recover their base RNNs by setting all gating weights to zero and only choose the online memory. Therefore, even in such cases, MC-enhanced variants cannot damage the performance and due to their benefit in other tasks, still they can be useful.
>
> Also, please note that, we acknowledge such cases and indeed there are room for future improvements as we have discussed in the paper as well. Here, we have focused on simple proof of concepts, and it would a very interesting future work to address the potential limitation of mean pooling methods. Please note that, the same argument is also valid for all other models based on mean pooling, such as Native Sparse Attention (Best Paper Awards in ACL 2025), and it would be interesting to explore alternative methods for mean pooling operator.

---

> > ### Author Rebuttal · Reviewer_kc99 · 2026-04-02
> >
> > The rebuttal is helpful and clarifies the intended scope of the paper. I agree that this line of work does not necessarily need to aim for state-of-the-art performance on all long-context benchmarks in order to be valuable. In my view, the community should encourage more exploration of alternative memory mechanisms for recurrent and linear-memory models, and this paper is clearly a thoughtful and careful effort in that direction.
> >
> > That said, I still think the paper would benefit from a clearer practical positioning. In particular, if the goal is not to outperform all efficient long-context alternatives, the paper should more explicitly explain when practitioners should prefer MC, what trade-offs it offers, and in what regimes it is most useful relative to other approaches. I see this as the main remaining gap after the rebuttal.
> >
> > Overall, I appreciate the care behind the work and the additional clarifications. Even though some of my concerns about positioning and evaluation remain only partially addressed, I view this as a well-motivated and earnest contribution that explores an important direction.

---

> > > ### Author Response · Authors · 2026-04-08
> > >
> > > Thank you for your time and the constructive feedback. We are glad that you found our work as a "a thoughtful and careful effort" in a right direction. We are also very encouraged that you are supporting exploration of alternative memory mechanisms.
> > >
> > > Following your suggestion, we will use additional space and clearly discuss and explain when practitioners should prefer MC. In particular: (1) MC is a compression and recurrence-based method, accordingly, it can be more helpful in applications and domains that are compressible and does not require direct lossless retrieval; (2) MC unifies recurrence and attention architectures, hence it provides a trade-off between lossless compression and efficiency, where using larger segments (having RNNs as its extreme case of 1 segment) would result in more efficient but more compression-based method; (3) When the task and domain require lossless retrieval, there are more suitable methods such as InfLLM-V2 and DSA mentioned by the reviewer;
> > >
> > > We hope the above responses have addressed your concern. We would really appreciate it if you could kindly consider re-evaluating the score of our paper. Once again, thank you very much for your time, for your encouraging feedback, and for engaging with us in the rebuttal phase.

---

### Official Review · Reviewer_vbm3 · 2026-03-11

**Soundness:** 3
**Presentation:** 3
**Significance:** 3
**Originality:** 2
**Overall Recommendation:** 4
**Confidence:** 4

**Summary:**

This work explores the idea of adding memory to RNNs.  The objective of this memory is to constrain the computational costs by a mechanism that “interpolates” between the linear cost on the sequence length of an RNN versus a quadratic cost of transformers.
By dividing the sequence into segments, the memory mechanism compresses previous inputs and queries within the current segment memory with a learning update. The learning update is recursive, following the ideas from Linear Attention (Katharopoulos at al., 2020) and related works.
In order to “recall” from memory, the output of the RNN aggregates the memories from previous segments, the current one, and the current query.
The paper proposes several aggregation methods: summation of memories, gated residual memory (that use input-dependent weights for a modulated summation),  and memory soup.  Additionally, the authors propose a more efficient method that uses Mixture of Experts (MoE) routing to select a subset of memories.

The different memory (caching) mechanisms described above are tested with various attention-based models (SWLA, DLA, and Titans).
Results show a small improvement in language modeling and common sense reasoning when using the gated residual memory (other mechanisms have smaller improvements). Needle-in-a-haystack results the GRM versions improve over the baselines, and better than the other mechanisms.
Additional results are presented for the in-context retrieval task, while memory helps improve the models, transformers are much better.
Similar results are shown for Multi-query associative recall and long context understanding tasks.
An ablation study shows the importance of each design option.

**Compliance With Llm Reviewing Policy:**

Affirmed.

**Final Justification:**

The paper adds to previous work in linear-like transformers with a memory cache.
It would be good to introduce the log-linear method more clearly, and to obtain some understandings of the inner workings of the MC proposal. The positioning of the paper could be improved, as also expressed by other reviewers.
The most performant method has quadratic complexity, which limits the significance of the solution.
Nonetheless, the work has some merit, so will keep the current score.

**Key Questions For Authors:**

* How does your method compare to other works that add memory to the networks?
* What would be a caching method to effectively reduce the complexity (beyond SSC), and how does it compare to the GRM results in the paper?
* How does the memory caching internally help improve the results?

**Limitations:**

The authors included a soft statement.
Some examples:
* Improved results in common sense reasoning can help reduce hallucinations. Still hallucinations exist in these models.
* Lower hardware barriers for training language models.
* Reduced training time potentially thanks to $O(N)$ (when lower than $O(L)$).

**Strengths And Weaknesses:**

Strengths:
* The work is clearly written and well organized. The methods section includes valuable discussions and insights that help understand the similarities and differences between the mechanisms.
* The experimental section is fairly well evaluated in a number of tasks, datasets and models with their memory caching mechanisms.  The ablation study helps understand the value of the mechanisms elements.
* The paper tackles an alternative view for the memory mechanism for recurrent attention models while considering the complexity costs. This is an active line of work for attention-like mechanisms.
* The proposed framework for combining the recurrent attention networks with memory caching is novel to the best of my knowledge. The methods are well articulated and clearly described.

Weaknesses:
* The results section shows similar results for many tasks. It would be nice to understand better the internal behavior of this methods, for example trying to shed some light on why memory caching, and in particular GRM, improves the results? Why are the improvements small (comparing to transformers on retrieval tasks) despite their memory caching?
* While the paper compares several recurrent attention networks among them with and without their proposed memory mechanisms, other techniques could have been brought to extend the methods as well and add more interesting comparisons. How does it compare with other memory-based techniques (for example like [1]), when applied to the recurrent attention networks?
* The authors claim that the memory caching technique with segment has $O(N)$ and N interpolates between 1 and L (sequence length). When, a memory caching mechanism with constant segments is used, we have that $O(N)=O(L)$. This leads to $O(L^2)$ similar to full attention. The only exception to this case is SSC variant. It could have been nice to propose an actual caching technique that grows at a slower rate than the sequence length and see if results still hold.


[1] Wu et al. Memorizing transformers. 2022

---

> ### Author Rebuttal · Authors · 2026-03-31
>
> Thank you for your detailed and constructive review. Following your suggestions, we have added additional experiments to address your questions, and as a result we believe these changes have strengthened our paper.
>
> > Why are the improvements small?
>
> **Response:** Please note that for some retrieval tasks, even a small compression can cause missing information and so we need full caching of raw tokens to have direct access to them, rather than caching their compressed memory states. Accordingly, despite improvement over recurrent models, still MC-enahnce methods might fall short on retrieval tasks, and we believe that one of the main motivations for MC. In fact, recurrent models are great for small context language modeling tasks as well as tasks that requires compression and are repeatedly reported to outperform Transformers in such tasks. On the other hand, for retrieval tasks, RNNs fall short compared to Transformers. This is mainly due to the fact that the context is compressed and some information is lost, as opposed to Transformers that cache every single token and so are capable of having direct access to the past data. MC provides an intermediate point, where the architecture perform caching on the states of the recurrent neural networks. Accordingly, it comes with both (dis)advantages: One main goal in our experiments is to show: (1) in short-context tasks, MC enhances the performance of its RNN baseline; and (2) in long-context and retrieval tasks it closes the performance gap with Transformers and again improves the performance of its RNN baseline.
>
> Please note that MC has provided significant improvement over the baseline recurrent neural networks. For retrieval tasks, MC shows **consistent** gain (most of the time 25%-40%) over the baselines across all datasets, and also show about 27% improvement on average. This is particularly clear on longer context: In NIAH tasks, we can even see about 400\% gain over DLA.
>
> > How does it compare with other memory-based techniques (for example like [1])?
>
> **Response:** Please note that the goal of our work with memory-based attentions are different. A group of studies have focused on improving the long-context capabilities of Transformers and make them more efficient by adding memory component. Another orthogonal group of studies have focused on the improving of recurrent models and make them more robust and performant for long-context data. While memory-based techniques mentioned by the reviewer have focused on the first direction, our work has targeted the second direction.
>
> We agree that there is an interesting direction to take inspiration from [1] and use use kNN-based retrieval for RNNs. However, this requires non-trivial amount of effort. This is mainly due to the fact that even retrieving relevant tokens based on kNN, we still need to compress them. Since this compression is based on the current token and cannot be done in an online manner, for each token, we need to first retrieve relevant past tokens using kNN and then go over them again to compress them. This causes computational overhead. Using kNN for attention, due to the parallelizability of computation can be very fast. Similarly, in memory caching, the cached memories are the same for all tokens (while they are combined in a data-dependent manner), and so we do not need to go over the past tokens again.
>
> We actually tried to incorporate such techniques into RNNs, and compare them, but unfortunately, it can make the model 20 times slower, and result in a huge computational cost for training.
>
> > Propose an actual caching technique that grows at a slower rate
>
> **Response:** Please note we already have this in the paper. In fact, although building on Log-linear attention, the Log-linear++ baseline is also one of our variants and its grow rate is theoretically slower than linear (i.e. it is logarithmic). This variant also provides improvement over the baseline. Also, please note that although theoretically the complexity of GRM can be seen as linearly growing memory (and so quadratic with sequence length), its grow rate can be 100 times slower than the actual global softmax attention.
>
> > How does the memory caching internally help improve the results?
>
> **Response:** Please note that in recurrent models, when compressing the context, the fixed-state size enforces more and more compression when the context grows, resulting in losing more details over time. Memory caching internally use the gates to directly retrieve some of the information from the past. Unfortunately, there is no revision opportunity, but following your suggestion, we have performed a new experiment visualizing the value of gates over the past cached memory states, showing how internally based on the token MC retrieve the data from the past. Mainly the results support the fact that based on the query, the model set non-zero weights to the past cached memory states, resulting in performance improvement.

---

> > ### Author Rebuttal · Reviewer_vbm3 · 2026-04-02
> >
> > I would like to thank the authors for the effort replying to my questions.
> >
> > The logarithmic approach is not properly defined in the main sections of the paper and the details of implementation are still vague. In addition, it would make sense to add the original Log-Linear attention approach as a baseline. Further, a comparison on the amount of memory used would help understand such comparison better. Could you further comment on the actual memory used for each of the MCs and what are the gains versus Transformers?
> >
> > I would appreciate it if the authors could share the results on their additional empirical analysis on the visualization of the internal workings of the MC-based models.

---

> > > ### Author Response · Authors · 2026-04-08
> > >
> > > **Response:** Thank you for your time and the constructive feedback.
> > >
> > > Following your suggestion, we will use the additional space and move all the details about the logarithmic method to the main body of the manuscript. Also, as you suggested, here we have provided the results for the original log-linear method:
> > >
> > > | Model | Avg. Acc. | Avg. NIAH | Memory Usage |
> > > |-|-|-|-|
> > > |Log-Linear | 56.4 | 77.8 | 27 GB |
> > > |Log-Linear++ | 57.2 | 78.6 | 27 GB |
> > > |GRM | 58.3 | 86.1 | 30 GB |
> > > |SSC | 57.6 | 81.2 | 30 GB |
> > >
> > > The variants of MC, including log-linear++, can outperform the original log-linear attention method. We will add the details of these results to the final version of the paper.
> > >
> > > We sincerely hope that these results resolve your concerns.

---

### Official Review · Reviewer_Aiit · 2026-03-12

**Soundness:** 4
**Presentation:** 3
**Significance:** 3
**Originality:** 3
**Overall Recommendation:** 5
**Confidence:** 4

**Summary:**

This paper introduces a Memory Caching technique intended to improve the memory capabilities of RNNs, while providing a controllable middle-ground of complexity between the linear time complexity of RNNs and quadratic complexity of Transformers. The framework functions by first segmenting input sequences and caching compressed memory checkpoints, which are later aggregated into a compressed history that influences later output computation. The paper proposes using four distinct aggregation methods: residual memory, gated residual memory, memory soup, and sparse selective caching, each of which draw inspiration from prior mechanisms introduced in RNNs and Transformers. The authors also touch upon the distinction of whether the memory caching process should be formulated as checkpointing (i.e. incorporating past history when caching a memory) versus independent memory compression (i.e. caching each memory without past input knowledge), and briefly explore the performance and efficiency disparity between the two approaches.

For evaluation, the authors provide a thorough set of results on models trained from scratch across a wide set of context window sizes, memory segment lengths, and with two model sizes (760M and 1.3B). They evaluate the perplexity and accuracy scores of these models on a collection of widely used, language modeling benchmarks (Wikitext, HellaSwag, WinoGrande, etc.), comparing them to several modern recurrent models and showing modest performance gains across all benchmarks. The authors also train recurrent models using their Memory Caching framework on several long context tasks (in-context recall, needle-in-haystack, and LongBench), comparing their evaluation results to a generic Transformer model. While the Memory Caching-enhanced RNNs' performance still lags behind that of the Transformer across these tasks, the Memory Caching techniques does show significant gains in task performance (especially when using Gated Residual Memories). The experiments also include an ablation study, confirming that each component of the gated residual memory and sparse selective caching aggregation mechanisms positively contribute to performance.

**Compliance With Llm Reviewing Policy:**

Affirmed.

**Final Justification:**

The work is a firmly situated in prior research and contributes a welcome addition to RNN-driven research, though as other reviewers pointed out, there could be more in-depth comparisons with these other methods. Nonetheless, the work has merit on its own and the authors answered question satisfactorily and included some additional experimentation.

**Key Questions For Authors:**

1. Was there any further analysis of the effect of segment size performed? The mention in the results is exceedingly small for what could be an important parameter for downstream performance.
2. The benchmark comparisons with the generic Transformer don't seem to provide any information on what this Transformer is for comparison. Could you provide more background details on this baseline for comparison?
2. You mention that the SSC approach is the best of both worlds in terms of efficiency, but it seems to perform well below GRM in the retrieval tasks. It seems odd that the SSC approach suffers especially on the Needle-in-a-Haystack task, where I would assume that past information would be most *irrelevant* to performing the task. Can you clarify/speculate on the discrepancy here?

**Limitations:**

Yes (but discussion of future improvements could be expanded).

**Strengths And Weaknesses:**

**Soundness:**

This work is firmly rooted in previously established work on modern recurrent models and attention mechanisms, and clearly cites prior work inspirations for the aggregation methods explored. The experiments and accompanying results all show a thorough and rigorous evaluation of the memory framework across a selection of tasks, and an ablation study is included to validate the positive impact of the aggregation mechanisms. One area that could be improved upon is the reporting of the effect of segment size and the analysis of checkpointing vs. independent memory compression. Segment size could have an enormous impact on downstream task performance, especially given different aggregation strategies, but this is left to only a brief mention in the results. The comparison between checkpointing and independent compressors is also only briefly explored. This is understandable given paper constraints, but this was highlighted as a significant question in the Methods and should at least have results reported in the supplementary material.

**Presentation:**

The paper is very well-structured in its narrative flow and covers each of the aggregation methods with clear and concise explanations. One comment is that the placement of Figures and Tables could be improved as reading the paper currently involves a lot of back and forth. For example, Figure 3 appears well before it is referenced in the text.

**Significance:**

While the final results for the Memory Caching-enhanced recurrent models still lag behind that of the Transformer by a significant margin, they do show a significant improvement over previous recurrent approaches. The improved performance and tunable complexity could have wider influence on future recurrent language modeling approaches.

**Originality:**

This work leverages previous developments in recurrent and attention mechanisms, but combines them in a novel fashion that is very well-articulated and has a clear logical through-line.

---

> ### Author Rebuttal · Authors · 2026-03-31
>
> Thank you for your detailed and constructive review. Following your suggestions, we have added additional experiments to address your questions, and as a result we believe these changes have strengthened our paper. Please find our detailed responses below:
>
> > The effect of segment size and the analysis of checkpointing vs. independent memory.
>
> **Response:** Regarding ablation on segment size, we agree that we have reported only a few number of cases, but please note that the result is conclusive as the pattern is: with increasing the segment size, the performance gets better until some point and then increasing the segment size start to damage the performance.
>
> However, following your suggestion, we have provided additional results on segment size and also a table to compare checkpointing vs. independent memory in terms of performance and efficiency:
>
>
> |Model | Performance | Efficiency
> | - | - | -|
> Checkpointing | 58.3 | 38 |
> independent | 57.2 | 44 |
>
> |Segment Size | Performance | Efficiency
> | - | - | -|
> C = 4 | 16.27 | 41 |
> C = 8 | 16.38 | 39 |
> C = 32 | 14.91 | 38 |
> C = 64 | 15.26 | 38 |
> C = 128 | 17.09 | 37 |
> C = 256 | 17.4 |  38 |
>
>
> > Figure 3 appears well before it is referenced in the text
>
> **Response:** Thank you very much, we will make sure to proof read the paper and fix all these issues in the final version of the paper.
>
>
> > Could you provide more background details on this baseline for comparison?
>
> **Response:** The baseline Transformer used in the paper is Llama Transformer; i.e., with RoPE and gated MLP blocks. It uses global causal softmax attention layers, meaning that each layer allows each token to attend to all past tokens.
>
> > SSC approach is the best of both worlds in terms of efficiency ... Can you clarify/speculate on the discrepancy here?
>
> **Response:** Thank you very much for mentioning that and sorry for the confusion. By mentioning having the "best of both worlds", we meant: SSC is a growing memory architecture (similar to Transformers), while the decoding cost is relatively constant per token (as in RNNs). Please note that, SSC is a subset of GRM, in the sense that GRM consider all past cached memories, while SSC chooses only a subset of them. Accordingly, GRM cannot work worse than SSC (it can assign zero weights to those memories that SSC is not choosing, and so recover SSC) and the results that SSC is below GRM is expected. However, on the other hand, SSC requires less computational cost and so it is a trade-off between SSC and GRM.
>
> We will make sure to clarify and revise the term of having best of both worlds. We didn't mean to say SSC is better in both effectiveness and efficiency, but we meant it has the good characteristics of both Transformers (growing memory) and RNNs (constant decoding cost per token).

---

> > ### Author Rebuttal · Reviewer_Aiit · 2026-04-03
> >
> > I appreciate the authors' response, especially the inclusion of the additional ablation experiment. I agree with other reviewers that some analysis of the method's efficiency could strengthen the paper, but I believe this work has merit for improving attention-free RNNs without comparison to other long-context/attention-based methods (although a deeper comparison would naturally be a welcome addition).

---

> > > ### Author Response · Authors · 2026-04-08
> > >
> > > Thank you very much for your time and constructive comments. We really appreciate your support of our work.

---

### Decision · Program_Chairs · 2026-04-30

**Decision:**

Accept (regular)

**Comment:**

This submission introduces Memory Caching, a simple and timely method for giving recurrent models growing memory. The article's main contribution is to bridge fixed-memory RNNs and quadratic-cost Transformers via cached memory checkpoints and flexible aggregation schemes. Reviewers found the paper technically solid, novel in formulation, and well grounded in prior work. A key strength is the broad empirical evaluation across language modeling, reasoning, retrieval, and long-context benchmarks.
The paper was also praised for its clear presentation and well-articulated methods section. The rebuttal addressed major questions, added ablations, and provided useful clarifications. Some efficiency comparisons could be stronger, but these concerns are limited, and there is a clear case for acceptance.